# TIGHTER PERFORMANCE THEORY OF FEDEXPROX

## ABSTRACT

We revisit FedExProx – a recently proposed distributed optimization method designed to enhance convergence properties of parallel proximal algorithms via extrapolation. In the process, we uncover a surprising flaw: its known theoretical guarantees on quadratic optimization tasks are no better than those offered by the vanilla Gradient Descent (GD) method. Motivated by this observation, we develop a novel analysis framework, establishing a tighter linear convergence rate for non-strongly convex quadratic problems. By incorporating both computation and communication costs, we demonstrate that FedExProx can indeed provably outperform GD, in stark contrast to the original analysis. Furthermore, we consider partial participation scenarios and analyze two adaptive extrapolation strategies – based on gradient diversity and Polyak stepsizes — again significantly outperforming previous results. Moving beyond quadratics, we extend the applicability of our analysis to general functions satisfying the Polyak-Łojasiewicz condition, outperforming the previous strongly convex analysis while operating under weaker assumptions. Backed by empirical results, our findings point to a new and stronger potential of FedExProx, paving the way for further exploration of the benefits of extrapolation in federated learning.

## 1 INTRODUCTION

Federated Learning (FL) (Konečný et al., 2016; McMahan et al., 2017) is a distributed machine learning paradigm where multiple edge devices collaboratively train a global model without the need for centralized data collection. Instead of transmitting raw data, each client computes updates based on their local datasets, and these are then aggregated at a central server to update the global model. This approach ensures that no sensitive information is shared with the orchestrating server or other clients, making it particularly useful in privacy-sensitive domains, such as healthcare (Rieke et al., 2020) and recommendation systems (Hard et al., 2018). Formally, FL solves the following optimization problem:

$$\min_{x \in \mathbb{R}^d} \left\{ f(x) := \frac{1}{n} \sum_{i=1}^{n} f_i(x) \right\}, \tag{1}$$

where $f_i : \mathbb{R}^d \to \mathbb{R}$ represents the empirical risk associated with the local dataset stored on client $i \in [n]$. Despite its advantages, FL introduces several nontrivial challenges, including *communication delays* caused by limited network bandwidth, and *partial participation* of clients due to random outages (Kairouz et al., 2021). Addressing these issues is essential to making FL efficient and reliable in real-world applications.

### 1.1 RELATED WORK

Let us briefly review the existing methods used to solve (1) in FL scenarios. For a detailed overview of the notation used, please refer to Table 1.

**GD.** The simplest and most naive approach is the *Gradient Descent* (GD) method (Nesterov, 2018), iterating $x_{k+1} = x_k - \gamma \nabla f(x_k) = x_k - \gamma \frac{1}{n} \sum_{i=1}^{n} \nabla f_i(x_k)$, where $x_0$ is a starting point and $\gamma$ is the stepsize. Let us introduce the key assumptions typically employed in the analysis of GD in the convex world.

**Assumption 1.1.** The functions $f_i$ are proper, closed and convex for all $i \in [n]$, and the function $f$ attains a minimum at some (potentially non-unique) point $x_* \in \mathbb{R}^d$.

**Assumption 1.2.** The function $f$ is $L$–smooth, i.e., $\|\nabla f(x) - \nabla f(y)\| \leq L \|x - y\| \; \forall x, y \in \mathbb{R}^d$. Additionally, each $f_i$, $i \in [n]$, is differentaible and $L_i$–smooth, with $L_{\max} := \max_{i \in [n]} L_i$.

Under these assumptions, classical optimization theory guarantees that GD solves problem (1) and finds $\bar{x} \in \mathbb{R}^d$ such that $f(\bar{x}) - f(x_*) \leq \varepsilon$ in $\mathcal{O}(LR^2/\varepsilon)$ iterations, where $R^2 := \|x_* - x_0\|^2$. In an FL setting, finding a solution to the main problem requires each client to participate in every communication round, and thus to communicate with the server

$$\mathcal{O}\left(\frac{LR^2}{\varepsilon}\right) \tag{2}$$

times, transmitting the local gradient $\nabla f_i(x_k)$ at each iteration $k$. While this approach is conceptually simple, its direct application in distributed environments leads to several challenges related to communication overhead and scalability.

**Partial Participation.** In real-world FL scenarios, not all clients remain active for the entire duration of training. Instead, a subset of clients is chosen at each iteration, typically based on practical factors such as device availability (e.g., battery life or network conditions) or statistical considerations (e.g., data heterogeneity) (Kairouz et al., 2021; Tyurin & Richtárik, 2023).

**Communication bottleneck.** Another key problem in distributed training is the *communication bottleneck* (Ramesh et al., 2021; Kairouz et al., 2021). Since the overall performance of a distributed algorithm is the product of the number of communication rounds $K$ needed to find a solution and the cost $C$ of one such round, there exist two main strategies to addressing this issue: $(i)$ minimizing $C$ and/or $(ii)$ minimizing $K$. Objective $(i)$ is typically achieved via compression of the information transmitted between the clients and the server (Beznosikov et al., 2020; Gruntkowska et al., 2024). Objective $(ii)$ can be accomplished by increasing the amount of local computation performed by clients between rounds, allowing for less frequent communication.

**FedAvg and modern local methods.** A common strategy for ensuring communication efficiency that falls into category $(ii)$ is *local training*. The idea is simple: allow clients to do more work before transmitting the results to the server. If executed with care, this approach leads to each communication round providing more "informative" updates, ultimately resulting in more effective changes to the global model. One of the most popular methods in this class is Federated Averaging (FedAvg), introduced by McMahan et al. (2017). Within this framework, a subset of clients is selected in each round to perform local training using a gradient-based method of their choice (e.g., GD). This is followed by an averaging step on the server, where the local updates are aggregated. Despite its popularity, the theoretical properties of FedAvg are somewhat limited (Khaled et al., 2019; Koloskova et al., 2020). A major issue is *client drift*, which can lead to slower convergence (Karimireddy et al., 2020). This problem can be mitigated using modern drift correction techniques (Karimireddy et al., 2020; Gorbunov et al., 2021; Mishchenko et al., 2022). One notable advancement in this area is FedExP, introduced by Jhunjhunwala et al. (2023), which extends FedAvg by leveraging extrapolation to better control server updates.

**FedProx and proximal methods.** All methods discussed so far assume that clients have access to the gradients $\{\nabla f_i\}$ of local functions. A recent line of *proximal methods* (Bertsekas, 2011; Ryu & Boyd, 2014; Khaled & Jin, 2022; Richtárik et al., 2024) relies on a different, more powerful oracle – the *proximal operator* $\text{prox}_{\gamma f} : \mathbb{R}^d \to \mathbb{R}^d$, defined by

$$\text{prox}_{\gamma f}(x) = \arg\min_{z \in \mathbb{R}^d} \left\{ f(z) + \frac{1}{2\gamma} \|z - x\|^2 \right\}. \tag{3}$$

Here, $f$ is a convex function and $\gamma$ plays a rule of a regularization parameter. It is important to note that computing $\text{prox}_{\gamma f}(\cdot)$ is an optimization problem on its own. However, the analyses of such methods essentially abstract away from the way of running the local operations, assuming that computing $\text{prox}_{\gamma f}(\cdot)$ is computationally cheap and can be performed using any solver, e.g., GD, Newton's method or LBFGS (Nesterov, 2018; Liu & Nocedal, 1989). There exist numerous proximal methods addressing the single-node case. A classic approach is the Proximal Point Method (PPM) by Rockafellar (1976), which iterates $x_{k+1} = \text{prox}_{\gamma_k f}(x_k)$. Stochastic versions of PPM have also been studied (Ryu & Boyd, 2014; Bianchi, 2016; Patrascu & Necoara, 2018). Recently, interest in stochastic proximal methods has surged, with promising results from several works (Condat & Richtárik, 2023; Traoré et al., 2024; Sadiev et al., 2024; Combettes & Madariaga, 2024). In the context of distributed learning, one relevant work by Li et al. (2020) builds on the principles of FedAvg

to introduce its proximal variant – FedProx. The method replaces local gradient-based updates with proximal operator computations, leading to the update rule

$$x_{k+1} = \frac{1}{n} \sum_{i=1}^{n} \text{prox}_{\gamma f_i}(x_k). \tag{4}$$

By leveraging proximal operators, FedProx addresses some of the limitations of traditional GD in FL. However, simply averaging local updates at the server remains suboptimal and can be improved.

**FedExProx.** A recent advancement, and the main focus of this paper, is FedExProx, introduced by Li et al. (2024). This method improves on FedProx by employing the technique of *extrapolation*[1] (Combettes, 1997; Necoara et al., 2019). The algorithm operates under the *interpolation regime*:

**Assumption 1.3.** There exists $x_* \in \mathbb{R}^d$ such that $\nabla f_i(x_*) = 0$ for all $i \in [n]$. We denote the set of such minimizers by $\mathcal{X}_* := \{x \in \mathbb{R}^d : \nabla f_i(x) = 0\}$ and the projection of $x$ onto $\mathcal{X}_*$ by $\Pi(x)$ .

This assumption is relatively mild and is often met in practice, e.g., when working with over-parameterized models (Ma et al., 2018). Instead of (4), FedExProx introduces the extrapolated update

$$x_{k+1} = x_k + \alpha_k \left( \frac{1}{n} \sum_{i=1}^{n} \text{prox}_{\gamma f_i}(x_k) - x_k \right), \tag{5}$$

where $\alpha_k$ serves as the *extrapolation parameter*. The algorithm is analyzed by reformulating step (5) in terms of *Moreau envelopes* (Moreau, 1965): the update rule can be expressed equivalently as

$$x_{k+1} = x_k - \alpha_k \gamma \frac{1}{n} \sum_{i=1}^{n} \nabla M_{f_i}^{\gamma}(x_k), \tag{6}$$

where the Moreau envelope $M_f^{\gamma} : \mathbb{R}^d \to \mathbb{R}$ of a convex function $f$ is defined as

$$M_f^{\gamma}(x) = \min_{z \in \mathbb{R}^d} \left\{ f(z) + \frac{1}{2\gamma} \|z - x\|^2 \right\} \quad \forall x \in \mathbb{R}^d. \tag{7}$$

Notably, when $\alpha_k = 1$, (5) reduces to (4). However, this choice is not optimal, and choosing $\alpha_k > 1$ yields better complexity. If $\alpha_k$ is set to be constant across iterations of the algorithm, then using the optimal value $\alpha_k \equiv \alpha = \frac{1}{\gamma L_\gamma} > 1$ guarantees that FedExProx converges after

$$\mathcal{O}\left( \frac{L_\gamma(1+\gamma L_{\max})R^2}{\varepsilon} \right) \tag{8}$$

communication rounds in the convex case, where $L_\gamma$ is the smoothness constant of the function $M^{\gamma}(x) := \frac{1}{n} \sum_{i=1}^{n} M_{f_i}^{\gamma}(x)$. Li et al. (2024) show that (8) is not larger than $\mathcal{O}\left( L_{\max}R^2/\varepsilon \right)$, and can be significantly smaller in many practical scenarios. Thus, FedExProx offers provably better iteration complexity compared to both FedProx and FedExP (Li et al., 2020; Jhunjhunwala et al., 2023).

## 2 CONTRIBUTIONS

**1.** Although the iteration complexity (8) of FedExProx outperforms both FedProx and FedExP, we find that it is *not superior* to that of the simplest baseline, GD, on quadratic optimization tasks (see Remark 3.3). This raises a critical question regarding the utility of employing the proximal oracle in the first place.

**2.** Motivated by these pessimistic findings, we revisit the analysis of FedProx and prove a significantly more optimistic iteration complexity result. We establish a linear convergence rate for solving non-strongly convex quadratic distributed optimization problems of the form (1) (see Theorem 4.1) and demonstrate that our new result does indeed lead to the conclusion that FedExProx can significantly outperform GD. Our analysis assumes a realistic model that accounts for both computation and communication times – critical factors in real-world distributed optimization. We show that the total *time complexity* of FedExProx is never worse than that of GD, and can be strictly better when communication time dominates computation time (see Theorem 4.3), which is typically the case in FL scenarios. This stands in stark contrast to the previous analysis by Li et al. (2024), underscoring the superiority of our new findings.

---

[1]Notably, the idea of extrapolation was explored by Jhunjhunwala et al. (2023) in developing FedExP.

**3.** To account for the stochastic setting that involves partial participation of clients, we complement the above results with client sampling (see Theorems 5.1, 5.2, 7.5 and F.1).

**4.** Beyond constant extrapolation, we establish two novel results (Theorems 6.1 and F.1) that incorporate smoothness-adaptive strategies, based on gradient diversity (FedExProx-GraDS), and Polyak stepsize (FedExProx-StoPS), again significantly improving upon the result previously established by Li et al. (2024) in the quadratic case.

**5.** We extend the analysis beyond quadratics to functions satisfying the Polyak-Łojasiewicz condition, obtaining a linear convergence rate (see Theorems 7.2 and 7.5). In contrast, Li et al. (2024) derived a linear rate under strong convexity for the function $f$. Our approach is not only more general, as it relies on weaker assumptions, but also demonstrates improved dependence on problem-specific constants. Additionally, we establish a result in the PŁ setting that accounts for inexact computations of the proximal mappings (Theorem E.1).

**6.** The theoretical findings are validated with empirical experiments, which demonstrate the robustness and applicability of our framework.

## 3   NOT BETTER THAN GD ON QUADRATICS

Let us take a closer look at the complexity result (8). This complexity depends on the regularization parameter $\gamma$ introduced in definition (3). Suppose for now that we solve (3) using GD. Then, the number of iterations needed to find $\mathrm{prox}_{\gamma f_i}(x)$ with accuracy $\varepsilon$ is

$$\mathcal{O}\left(\frac{L_i + 1/\gamma}{1/\gamma} \log \frac{1}{\varepsilon}\right) = \mathcal{O}\left((\gamma L_i + 1) \log \frac{1}{\varepsilon}\right),$$

since the function $f_i(z) + \frac{1}{2\gamma}\|z - x\|^2$ is $(L_i + 1/\gamma)$–smooth and $1/\gamma$–strongly convex. Hence, $\gamma$ controls the difficulty of calculating the proximal operator: the larger it is, the more difficult the problem. Therefore, to accelerate local computations, one would prefer to choose $\gamma$ as small as possible. We can formalize this intuition by letting $\pi(\gamma)$ be the time per one iteration of FedExProx and making the following reasonable assumption:

**Assumption 3.1.** The time complexity $\pi(\gamma)$ of a FedExProx step is a non-decreasing function of $\gamma$.

Combining it with the iteration complexity result (8) established by Li et al. (2024), the total time required by FedExProx with $\alpha_k \equiv 1/\gamma L_\gamma$ to find $\bar{x}$ such that $\mathbb{E}[f(\bar{x})] - f(x_*) \le \varepsilon$ is

$$T(\gamma) := \pi(\gamma) \times \frac{L_\gamma(1+\gamma L_{\max})R^2}{\varepsilon}.$$

With this definition in place, we can now translate the main result of Li et al. (2024) (Theorem 1) to quadratic optimization problems, yielding the following pessimistic outcome:

**Theorem 3.2.** *Let Assumptions 1.3 and 3.1 hold. Consider solving a non-strongly convex quadratic optimization problem of the form* (1), *where* $f_i(x) = \frac{1}{2}x^\top \mathbf{A}_i x - b_i^\top x$ *for all* $i \in [n]$, *with* $\mathbf{A}_i \in \mathrm{Sym}_+^d$ *and* $b_i \in \mathbb{R}^d$. *Then the assumptions of Theorem 1 by Li et al. (2024) hold, and*

$$T(\gamma) = \pi(\gamma) \times \frac{L_\gamma(1+\gamma L_{\max})R^2}{\varepsilon} \ge \pi(0) \times \frac{LR^2}{\varepsilon} \tag{9}$$

*for all* $\gamma > 0$. *Moreover, when* $\gamma \to 0$, *then* $\pi(\gamma) \times \frac{L_\gamma(1+\gamma L_{\max})R^2}{\varepsilon} \to \pi(0) \times \frac{LR^2}{\varepsilon}$, *and* FedExProx *effectively reduces to* GD.

*Remark* 3.3. In light of Theorem 3.2, GD performs no worse than FedExProx by Li et al. (2024), even if the time complexity $\pi(\gamma)$ of one FedExProx step does not depend on $\gamma$. Indeed, it holds that

$$\frac{L_\gamma(1+\gamma L_{\max})R^2}{\varepsilon} \ge \frac{LR^2}{\varepsilon}$$

for all $\gamma > 0$ (see the proof of Theorem 3.2).

These results, based on the original analysis of FedExProx, lead us to a rather disappointing conclusion: from a theoretical time complexity perspective, the optimal strategy appears to be using vanilla GD, disregarding proximal oracles entirely. The question that remains is: is this an inherent limitation of the method itself or merely a consequence of suboptimal analysis by Li et al. (2024)?

## 4 TIGHTER AND MORE OPTIMISTIC RESULTS

Are we truly constrained to GD, or is there room for improvement? It turns out that a tighter iteration complexity bound for FedExProx can be derived, revealing significantly more optimistic results. The following theorem presents a refined analysis, providing a new, improved complexity result for FedExProx and shedding light on its true capabilities.

**Theorem 4.1.** *Fix any $\gamma > 0$ and consider solving non-strongly convex quadratic optimization problem* (1) *where $f_i(x) = \frac{1}{2} x^\top \mathbf{A}_i x - b_i^\top x$ for all $i \in [n]$, with $\mathbf{A}_i \in \mathrm{Sym}_+^d$ and $b_i \in \mathbb{R}^d$. Under Assumption 1.3, FedExProx with $\alpha = 1/\gamma L_\gamma$ finds $\bar{x}$ such that $\mathbb{E}[f(\bar{x})] - f(x_*) \leq \varepsilon$ after*

$$\mathcal{O}\left(\frac{L_\gamma}{\mu_\gamma^+} \log \frac{1}{\varepsilon}\right) \tag{10}$$

*iterations, where $L_\gamma$ is a smoothness constant of $M^\gamma$ and $\mu_\gamma^+$ is the smallest non-zero eigenvalue of the matrix $\nabla^2 M^\gamma$.*

*Remark* 4.2. To the best of our knowledge, under the assumptions of Theorem 4.1, vanilla GD requires

$$\mathcal{O}\left(\frac{L}{\mu^+} \log \frac{1}{\varepsilon}\right) \tag{11}$$

iterations to solve the quadratic optimization problem, where $\mu^+$ is the smallest non-zero eigenvalue of the matrix $\mathbf{A} = \frac{1}{n} \sum_{i=1}^n \mathbf{A}_i$ (Richtárik & Takáč, 2020).

Let us now demonstrate that our new result does indeed provide a tighter bound, leading to the conclusion that FedExProx can in fact outperform GD.

### 4.1 ONE STEP TIME COMPLEXITY $\pi(\gamma)$ IS $\mu + \tau (\gamma L_{\max} + 1)$.

As introduced in Section 1, the time per one global iteration of FedExProx has two main sources:

1. **Local computation:** In large-scale problems, each step (5) requires clients to compute $\mathrm{prox}_{\gamma f_i}(x_k)$ iteratively. One of the simplest solvers is GD, which returns a solution of subproblem $i$ after $\widetilde{\mathcal{O}}(\gamma L_i + 1)$ local iterations[2] (see Section 3). If each gradient calculation takes $\tau$ seconds, the total time required for all clients to calculate $\mathrm{prox}_{\gamma f_i}(x_k)$ is $\widetilde{\mathcal{O}}(\tau \times (\gamma L_{\max} + 1))$, since the process is gated by the "slowest" client, associated with the problem with the highest smoothness constant.

2. **Communication:** Once the local computations are completed, clients must communicate their results before the server can execute the global step (5). We assume this communication takes $\mu$ seconds, which can be huge in FL environments (Kairouz et al., 2021).

Consequently, the total time per global iteration, $\pi(\gamma)$, is proportional to $\mu + \tau (\gamma L_{\max} + 1)$ (thus satisfying Assumption 3.1), and the total time complexity of FedExProx is given by

$$T_\mu(\gamma) := \widetilde{\mathcal{O}}\left((\mu + \tau (\gamma L_{\max} + 1)) \times \frac{L_\gamma}{\mu_\gamma^+}\right). \tag{12}$$

Note that the total time complexity of GD is

$$T_{\mathsf{GD}} := T_\mu(0) = \widetilde{\mathcal{O}}\left((\mu + \tau) \times \frac{L}{\mu^+}\right).$$

Our next goal is to determine the value of $\gamma$ that minimizes $T_\mu(\gamma)$, with the hope that this time the optimal choice does not result in $\gamma \to 0$. This is indeed the case whenever $\mu > \tau$.

**Theorem 4.3.** *Consider the non-strongly convex quadratic optimization problem from Theorem 4.1. Up to a constant factor, the time complexity* (12) *is minimized by $\gamma \in \left[\frac{1}{\max_{i \in [n]} \lambda_{\max}(\mathbf{A}_i)}, \min\left\{\frac{\frac{\mu}{\tau} - 1}{\max_{i \in [n]} \lambda_{\max}(\mathbf{A}_i)}, \frac{1}{\min_{i \in [n]} \lambda_{\min}^+(\mathbf{A}_i)}\right\}\right]$ if $\frac{\mu}{\tau} \geq 2$ and by $\gamma \in \left[0, \max\left\{0, \min\left\{\frac{\frac{\mu}{\tau} - 1}{\max_{i \in [n]} \lambda_{\max}(\mathbf{A}_i)}, \frac{1}{\min_{i \in [n]} \lambda_{\min}^+(\mathbf{A}_i)}\right\}\right\}\right]$ if $\frac{\mu}{\tau} < 2$, and $T_\mu(\gamma) \leq T_{\mathsf{GD}}$.*

---

[2]Alternatively, an accelerated method could be employed, reducing the iteration count to $\widetilde{\mathcal{O}}(\sqrt{\gamma L_i + 1})$. Our analysis can be extended to accommodate these faster solvers as well.

The theorem above establishes that the total time complexity of FedExProx is no worse than that of GD, and can be strictly better when communication time dominates computation time, highlighting the superiority of our new analysis. Notably, this was not the case for the previous analysis by Li et al. (2024) (Theorem 3.2). The potential for our complexity to significantly outperform that of GD is illustrated in the following simple example, where $T_\mu(\gamma) \ll T_{\mathsf{GD}}$ for some $\gamma > 0$.

*Example* 4.4. Let the matrices $\{\mathbf{A}_i\}$ be diagonal, i.e., $\mathbf{A}_i = \operatorname{diag}(a_{i1}, \ldots, a_{id})$ for all $i \in [n]$, and $a_{ij} > 0$ for all $i \in [n], j \in [d]$. Then, according to (27) and (28), we get

$$\frac{L_\gamma}{\mu_\gamma^+} = \frac{\lambda_{\max}(\gamma)}{\lambda_{\min}^+(\gamma)} = \frac{\max\limits_{j \in [d]} \sum\limits_{i=1}^{n} \frac{\gamma a_{ij}}{1 + \gamma a_{ij}}}{\min\limits_{j \in [d]} \sum\limits_{i=1}^{n} \frac{\gamma a_{ij}}{1 + \gamma a_{ij}}},$$

where $\lambda_{\max}(\gamma)$ and $\lambda_{\min}^+(\gamma)$ are the largest and the smallest non-zero eigenvalues of the matrix $\mathbf{M} := \frac{1}{n} \sum_{i=1}^{n} \frac{1}{\gamma}(\mathbf{I} - (\gamma \mathbf{A}_i + \mathbf{I})^{-1})$, respectively (see Section C). Taking $\gamma = 1/\min_{i \in [n], j \in [d]} a_{ij}$ and using the fact that $1 \geq \frac{x}{1+x} \geq \frac{1}{2}$ for all $x \geq 1$, we can conclude that $\frac{L_\gamma}{\mu_\gamma^+} \leq 2$. Thus,

$$T_\mu(\gamma) = \widetilde{\mathcal{O}}\left(\mu + \tau \frac{\max\limits_{i \in [n], j \in [d]} a_{ij}}{\min\limits_{i \in [n], j \in [d]} a_{ij}}\right).$$

Suppose that communication is slow. Formally, let $\mu \geq \tau \max_{i \in [n], j \in [d]} a_{ij} / \min_{i \in [n], j \in [d]} a_{ij}$. Then $T_\mu(\gamma) = \widetilde{\mathcal{O}}(\mu)$, while the total time complexity of GD is

$$T_{\mathsf{GD}} = \widetilde{\Omega}\left(\mu \times \frac{L}{\mu^+}\right) = \widetilde{\Omega}\left(\mu \times \frac{\max\limits_{j \in [d]} \sum\limits_{i=1}^{n} a_{ij}}{\min\limits_{j \in [d]} \sum\limits_{i=1}^{n} a_{ij}}\right),$$

which is at least $\max\limits_{j \in [d]} \sum_{i=1}^{n} a_{ij} / \min\limits_{j \in [d]} \sum_{i=1}^{n} a_{ij}$ times worse!

A similar improvement can be observed in the general case. However, the derivation is significantly more complex, so to maintain clarity in the presentation, the example focuses on diagonal matrices.

## 5 PARTIAL PARTICIPATION

Thus far, we have concentrated on the full participation scenario. However, as outlined in Section 1.1, practical FL settings often involve only a subset of clients participating in each training round. To address this, we supplement our theory with a convergence result in the stochastic setting. For illustration, we consider nice sampling (see Section A.2), where at each iteration, a subset $\mathcal{S}_k \subseteq [n]$ of clients is selected uniformly at random from all subsets of size $S$. Although we use this sampling strategy as an example, other client selection methods can also be employed.

In this context, we can formulate the following complexity result.

**Theorem 5.1.** *Fix any $\gamma > 0$ and consider solving non-strongly convex quadratic optimization problem* (1) *where $f_i(x) = \frac{1}{2} x^\top \mathbf{A}_i x - b_i^\top x$ for all $i \in [n]$, with $\mathbf{A}_i \in \operatorname{Sym}_+^d$ and $b_i \in \mathbb{R}^d$. Under Assumption 1.3, FedExProx with nice sampling (Algorithm 2) with $\alpha = 1/\gamma L_{\gamma,S}$ finds $\bar{x}$ such that $\mathbb{E}[f(\bar{x})] - f(x_*) \leq \varepsilon$ after*

$$\mathcal{O}\left(\frac{L_{\gamma,S}}{\mu_\gamma^+} \log \frac{1}{\varepsilon}\right) \tag{13}$$

*iterations, where $L_{\gamma,S} := \frac{n-S}{S(n-1)} \frac{L_{\max}}{1 + \gamma L_{\max}} + \frac{n(S-1)}{S(n-1)} L_\gamma$, $L_\gamma$ is the smoothness constant of $M^\gamma$ and $\mu_\gamma^+$ is the smallest non-zero eigenvalue of the matrix $\nabla^2 M^\gamma$.*

Assuming the same time complexity model as in Section 4.1, the total time complexity of FedExProx with nice sampling is given by

$$T_\mu(\gamma, S) := \widetilde{\mathcal{O}}\left((\mu + \tau(\gamma L_{\max} + 1)) \times \frac{L_{\gamma,S}}{\mu_\gamma^+}\right). \tag{14}$$

As it turns out, the optimal stepsize for the stochastic setting aligns with the one used in the deterministic scenario (see Theorem 4.3).

**Theorem 5.2.** *Up to a constant factor, the time complexity* (14) *is minimized by* $\gamma \in \left[\frac{1}{\max_{i \in [n]} \lambda_{\max}(\mathbf{A}_i)}, \min\left\{\frac{\frac{\mu}{\tau}-1}{\max_{i \in [n]} \lambda_{\max}(\mathbf{A}_i)}, \frac{1}{\min_{i \in [n]} \lambda_{\min}^+(\mathbf{A}_i)}\right\}\right]$ *if* $\frac{\mu}{\tau} \geq 2$ *and by* $\gamma \in \left[0, \max\left\{0, \min\left\{\frac{\frac{\mu}{\tau}-1}{\max_{i \in [n]} \lambda_{\max}(\mathbf{A}_i)}, \frac{1}{\min_{i \in [n]} \lambda_{\min}^+(\mathbf{A}_i)}\right\}\right\}\right]$ *if* $\frac{\mu}{\tau} < 2$.

Consequently, the conclusions from the previous section apply equally to the partial participation scenario: FedExProx performs at least as well as GD and can be strictly better when communication time exceeds computation time.

# 6 ADAPTIVITY

Next, we explore adaptive extrapolation strategies introduced by Horváth et al. (2022): *gradient diversity* (GraDS) and a variant of the classical *Polyak stepsize* (StoPS), both of which were later adapted for proximal methods by Li et al. (2024). As in the constant extrapolation case, we refine the analysis by Li et al. (2024) for quadratic problems. Although Theorem 6.1 refers to the full participation setting, the approach can be extended to the stochastic case (see Section F.2).

**Theorem 6.1.** *Fix any* $\gamma > 0$ *and consider solving non-strongly convex quadratic optimization problem* (1), *where* $f_i(x) = \frac{1}{2}x^\top \mathbf{A}_i x - b_i^\top x$ *for all* $i \in [n]$, *with* $\mathbf{A}_i \in \mathrm{Sym}_+^d$ *and* $b_i \in \mathbb{R}^d$. *Let Assumption 1.3 hold and consider two adaptive extrapolation strategies:*

*1. (FedExProx-GraDS) Set*

$$\alpha_k = \alpha_k^{\mathsf{GraDS}}(x_k) := \frac{\frac{1}{n}\sum_{i=1}^n \left\|\nabla M_{f_i}^\gamma(x_k)\right\|^2}{\left\|\frac{1}{n}\sum_{i=1}^n \nabla M_{f_i}^\gamma(x_k)\right\|^2} \geq 1.$$

*Then, the iterates of Algorithm 1 satisfy*

$$\|x_K - \Pi(x_K)\|^2 \leq \left(1 - \min_{k=0,\dots,K-1} \alpha_k \gamma \frac{2+\gamma L_{\max}}{1+\gamma L_{\max}}\mu_\gamma^+\right)^K \|x_0 - \Pi(x_0)\|^2. \quad (15)$$

*2. (FedExProx-StoPS) Set*

$$\alpha_k = \alpha_k^{\mathsf{StoPS}}(x_k) := \frac{\frac{1}{n}\sum_{i=1}^n \left(M_{f_i}^\gamma(x_k) - \inf M_{f_i}^\gamma\right)}{\gamma\left\|\frac{1}{n}\sum_{i=1}^n \nabla M_{f_i}^\gamma(x_k)\right\|^2} \geq \frac{1}{2\gamma L_\gamma}.$$

*Then, the iterates of Algorithm 1 satisfy*

$$\|x_K - \Pi(x_K)\|^2 \leq \left(1 - \frac{3}{2}\min_{k=0,\dots,K-1} \alpha_k \gamma \mu_\gamma^+\right)^K \|x_0 - \Pi(x_0)\|^2. \quad (16)$$

*Remark* 6.2. Similar to the observations by Li et al. (2024), we see that, unlike in the constant extrapolation case (see Theorem 4.1), FedExProx with adaptive extrapolation benefits from *semi-adaptivity* to the smoothness constant. Specifically, it converges for any $\gamma > 0$, and since $\gamma\mu_\gamma^+$ is bounded above (see (28)), it suffices to choose a large enough $\gamma$ to achieve the optimal performance.

*Remark* 6.3. Bounding $\min_{k=0,\dots,K-1} \alpha_k$ by $\frac{1}{2\gamma L_\gamma}$, inequality (16) implies that

$$\|x_K - \Pi(x_K)\|^2 \leq \left(1 - \frac{3}{4}\frac{\mu_\gamma^+}{L_\gamma}\right)^K \|x_0 - \Pi(x_0)\|^2, \quad (17)$$

and hence Theorem 6.1 guarantees convergence of FedExProx-StoPS in

$$\mathcal{O}\left(\frac{L_\gamma}{\mu_\gamma^+}\log\frac{1}{\varepsilon}\right)$$

iterations, regardless of the choice of the stepsize $\gamma$. This matches the guarantee from Theorem 4.1, but without requiring prior knowledge of the optimal extrapolation parameter $\alpha = 1/\gamma L_\gamma$. The benefit comes with the trade-off of needing to know the minimum of the average of Moreau envelopes.

# 7 BETTER THEORY WITH POLYAK-ŁOJASIEWICZ CONDITION

The result from Section 4 can be extended to solving problem (1) for general functions that satisfy the Polyak-Łojasiewicz (PŁ) condition.

**Assumption 7.1.** The function $M^\gamma$ satisfies PŁ-condition, i.e., there exists $\mu_\gamma^+$ such that

$$\frac{1}{2} \|\nabla M^\gamma(x)\|^2 \geq \mu_\gamma^+ \left(M^\gamma(x) - M^\gamma(x_*)\right)$$

for all $x \in \mathbb{R}^d$.

This assumption holds with $\mu_\gamma^+ \geq \mu^+/(1 + \gamma L_{\max})$ if the function $f$ satisfies PŁ condition with a constant $\mu^+$ (see Lemma A.11). However, the choice $\mu_\gamma^+ = \mu^+/(1 + \gamma L_{\max})$ is loose and leads to the problems described in Theorem 3.2. With this assumption in place, we can present the theorem.

**Theorem 7.2.** *Let Assumptions 1.1, 1.2, 1.3, and 7.1 hold. For all $\gamma > 0$,* FedExProx *(Algorithm 1) with $\alpha = 1/\gamma L_\gamma$ finds $\bar{x}$ such that $\mathbb{E}[f(\bar{x})] - f(x_*) \leq \varepsilon$ in*

$$\mathcal{O}\left(\frac{L_\gamma}{\mu_\gamma^+} \log \frac{1}{\varepsilon}\right)$$

*iterations, where $L_\gamma$ is a smoothness constant of $M^\gamma$ and $\mu_\gamma^+$ is the PŁ constant.*

*Remark* 7.3. In practice, obtaining an exact solution to the local problems is often infeasible. Instead, we can only calculate an inexact proximal operator that approximates the desired quantity within a certain accuracy budget. To accommodate this, we extend the above analysis to the scenario where the clients can only compute updates $\text{prox}^\delta_{\gamma f_i}(x)$ such that $\|\text{prox}^\delta_{\gamma f_i}(x) - \text{prox}_{\gamma f}(x)\|^2 \leq \delta$ (see Section E).

*Remark* 7.4. Li et al. (2024) also establish a linear rate for FedExProx. However, their result relies on the assumption that the function $f$ is strongly convex, and hence that (1) has a unique solution. In contrast, our theorem is more general, as it allows for the possibility of multiple solutions.

In the partial participation scenario, we can establish the following result.

**Theorem 7.5.** *Let Assumptions 1.1, 1.2, 1.3, and 7.1 hold. For all $\gamma > 0$,* FedExProx *with nice sampling (Algorithm 2) and $\alpha = 1/\gamma L_\gamma$ finds $\bar{x}$ such that $\mathbb{E}[f(\bar{x})] - f(x_*) \leq \varepsilon$ in*

$$\mathcal{O}\left(\frac{L_{\gamma,S}}{\mu_\gamma^+} \log \frac{1}{\varepsilon}\right)$$

*iterations, where $L_{\gamma,S} := \left(\frac{n-S}{S(n-1)}\frac{L_{\max}}{1+\gamma L_{\max}} + \frac{n(S-1)}{S(n-1)}L_\gamma\right)$, $L_\gamma$ is a smoothness constant of $M^\gamma$ and $\mu_\gamma^+$ is the PŁ constant.*

Note that the complexities in Theorems 7.2 and 7.5 are entirely analogous to those in Theorems 4.1 and 5.1, with the only distinction being the substitution of the smallest non-zero eigenvalue of the function $M^\gamma$ with the PŁ constant.

## 7.1 WHY DO WE GET A TIGHTER ANALYSIS?

To understand why the new analysis yields stronger guarantees, we need to examine the reasoning behind Theorem 3.2. When proving the result, we noticed that the main reason why (9) was true was the dependence of the complexity on the $L_{\max}$ factor. At the same time, the complexity of GD depends only on $L$, and there exist many examples when $L_{\max} \gg L$. The question was: why does the original analysis of FedExProx involve $L_{\max}$, and can one circumvent this dependency? The reason behind it is the reliance of the proofs by Li et al. (2024) on Lemma A.9, which establishes the inequality $M^\gamma(x) - M^\gamma(x_*) \geq \frac{1}{1+\gamma L_{\max}}(f(x) - f(x_*))$ for all $x \in \mathbb{R}^d$. This inequality appears to be essential for deriving the convergence result $\mathbb{E}[f(\bar{x})] - f(x_*) \leq \varepsilon$ (where $\bar{x} \in \mathbb{R}^d$ is the output of FedExProx). Our main idea is to instead obtain convergence in terms of the distance between $\bar{x}$ and $x_*$, i.e., to establish that $\mathbb{E}[\|\bar{x} - x_*\|^2] \leq \varepsilon$, and then translate it to $\mathbb{E}[f(\bar{x})] - f(x_*) \leq \varepsilon$ using the $L$–smoothness of the function $f$. In this way, one can avoid using Lemma A.9 when dealing with quadratic functions or problems satisfying the PŁ condition, ultimately obtaining much more favourable convergence guarantees.

The story behind our theoretical improvements is no less important than the improvements themselves, as it offers valuable insights and can be instructive for future research.

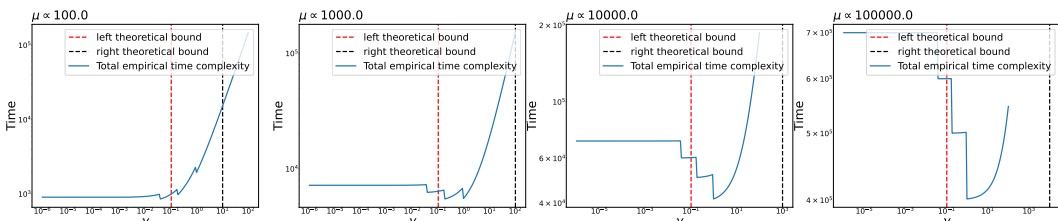

Figure 1: Empirical time complexities of FedExProx on a quadratic optimization task.

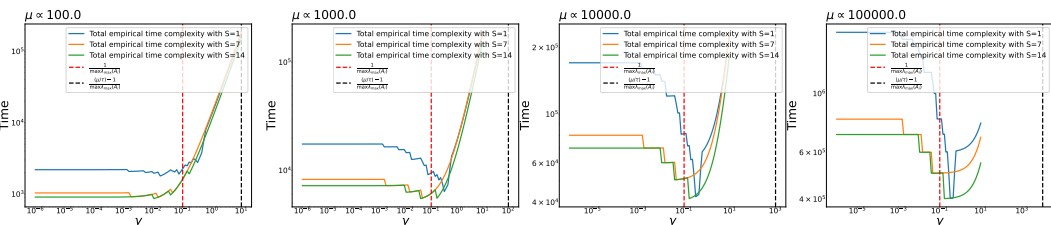

Figure 2: Empirical time complexities of FedExProx with partial client participation on a quadratic optimization task for $S \in \{1, 7, 14\}$ clients participating in each round.

## 8 EXPERIMENTS

To verify our theoretical results, we assess the empirical time complexity of FedExProx as a function of $\gamma$. We consider two types of optimization problems: synthetic quadratic optimization tasks and classification problems with a smooth hinge loss function. In both cases, local updates are computed using GD. Below, we first outline the methodology common to both experiments, followed by a detailed description of the specific settings and their corresponding results.

**Total empirical time complexity.** In accordance with the setup described in Section 4.1, we assume that one local iteration of GD takes $\tau$ seconds. Thus, the time needed by worker $i$ to find $\text{prox}_{\gamma f_i}(x_k)$ at global iteration $k$ is proportional to $\tau \times n_{ik}$, where $n_{ik}$ is the number of GD iterations needed to find $\text{prox}_{\gamma f_i}(x_k)$ to a given small fixed accuracy. In the full participation case, the total empirical time complexity is given by

$$\sum_{k=0}^{K-1} \left( \mu + \tau \max_{i \in [n]} n_{ki} \right) \tag{18}$$

where $K$ is the number of global iterations needed for FedExProx to converge to the desired accuracy and $\mu$ is the communication cost. Notice that we take the maximum over $n_{\cdot i}$ because the overall time is determined by the slowest client. One of experimental tasks is to verify that this empirical complexity aligns with the theoretical complexity derived in (14) (see Section 8.2). In experiments involving client sampling, we consider the complexity $\sum_{k=0}^{K-1} (\mu + \tau \max_{i \in \mathcal{S}_k} n_{ki})$ instead, where $\mathcal{S}_k$ is the random subset of clients sampled at global iteration $k$.

### 8.1 EXPERIMENTS WITH QUADRATIC OPTIMIZATION PROBLEMS

In the first set of experiments, we consider quadratic optimization problems of the form $f(x) = \frac{1}{n} \sum_{i=1}^{n} \frac{1}{2} x^\top \mathbf{A}_i x$, where $\mathbf{A}_i \in \text{Sym}_+^7$, $i \in [n]$ are random positive semidefinite matrices with the smallest eigenvalue equal to zero. Each worker $i$ computes the proximal mappings using GD with a stepsize of $1/\bar{L}_i$, where $\bar{L}_i = \lambda_{\max}(\mathbf{A}_i) + 1/\gamma$ is the smoothness constant of the local subproblem (3). The extrapolation parameter $\alpha$ is set to its optimal value from Theorem 4.1, using an explicitly computed value of $L_\gamma$. The data is distributed across $n = 14$ workers.

The results are presented in Figure 1. The dashed lines represent the theoretical bounds from Theorem 4.3, within which the optimal $\gamma$ is expected to lie. One can see that when $\mu$ is relatively small ($\mu \propto 100$), the best choice of $\gamma$ is near 0. However, as the communication cost $\mu$ increases, the best $\gamma$ shifts to values greater than 0.1. A distinctive U-shape emerges, indicating the nontrivial optimal choice of $\gamma$. The results for partial participation from Figure 2 are qualitatively the same. These observations are fully consistent with our theoretical predictions.

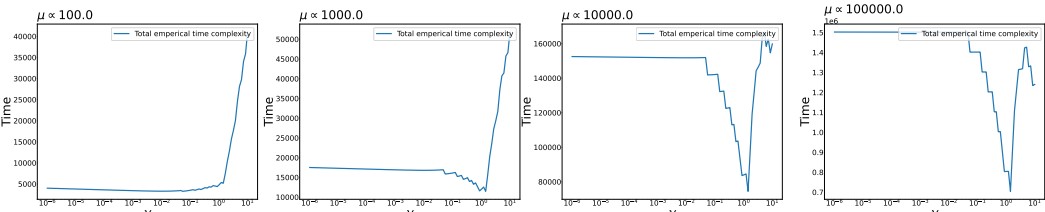

Figure 3: Empirical time complexities of `FedExProx` on a classification task with smooth hinge loss.

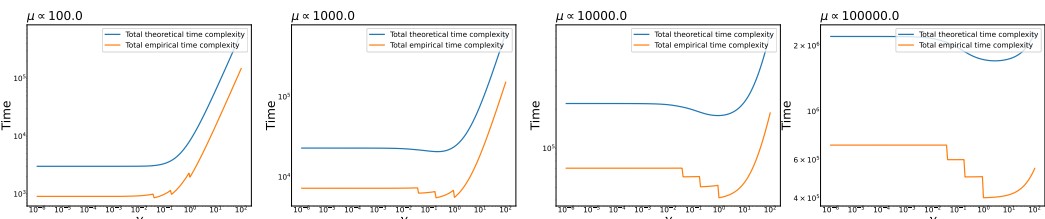

Figure 4: Comparison of theoretical time complexity (12) and empirical time complexity (18).

## 8.2 EXPERIMENTS WITH SMOOTH HINGE LOSS

We now turn to a classification problem with linearly separable data $\{(x_{ij}, y_{ij})\}_{i \in [n], j \in [m]}$, ensuring the interpolation regime. The function $f : \mathbb{R}^d \to \mathbb{R}$ is defined as

$$f(w) = \frac{1}{n} \sum_{i=1}^{n} f_i(w) := \frac{1}{n} \sum_{i=1}^{n} \frac{1}{m} \sum_{j=1}^{m} \ell_{ij}(w),$$

where $(x_{ij}, y_{ij}) \in \mathbb{R}^d \times \{-1, 1\}$ for all $i \in [n], j \in [m]$ and $\ell_{ij}$ is the smooth hinge loss, defined by

$$\ell_{ij}(w) = \begin{cases} 0, & \text{if } y_{ij} w^T x_{ij} \geq 1, \\ \frac{1}{2}(1 - y_{ij} w^T x_{ij})^2, & \text{if } 0 < y_{ij} w^T x_{ij} < 1, \\ 1 - y_{ij} w^T x_{ij}, & \text{if } y_{ij} w^T x_{ij} \leq 0. \end{cases}$$

Each client $i \in [n]$ computes the local proximal mappings using GD with stepsize $1/\bar{L}_i$, where $\bar{L}_i = \frac{1}{m} \sum_{j=1}^{m} \|x_{ij}\|^2$ is an upper bound on the smoothness constant of the function $f_i$. The extrapolation parameter $\alpha$ is determined via grid search within the theoretical range given by Lemma A.7. We use $n = 4$ workers, each holding $m = 4$ datapoints, and the number of parameters is $d = 3$.

The results are presented in Figure 3. As in Section 8.1, these experiments also suggest a non-zero optimal choice of $\gamma$ when communication is slow. Indeed, the values of $\gamma$ minimizing the U-shapes in the last three plots are around 1. These experiments provide further support for our theory.

**Comparing theoretical and empirical time complexities.** Lastly, we compare the theoretical and empirical time complexities (12) and (18). Both are plotted on the same scale in Figure 4. We see that our theoretical model reflects the empirical dependencies (up to a multiplicative factor).

## 9 CONCLUSION

In this work, we revisit an empirically promising extrapolated parallel proximal method `FedExProx`. Motivated by the limitations of its original overly pessimistic analysis, we develop a novel analytical framework for non-strongly convex quadratic and PŁ cases. Our analysis significantly improves upon the previous results, offering insights into the true method's performance. While our work focuses on the relatively simple case of quadratic problems, the experimental results indicate that the phenomena we describe extend beyond this setting. These findings suggest that the benefits of extrapolation in FL hold in broader optimization scenarios, motivating future research to rigorously establish such extensions. We believe that our work provides a foundational stepping stone toward better understanding the role of extrapolation in FL and can inspire further exploration to relax the assumptions made here, ultimately advancing the theory and practice of federated optimization.

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

CONTENTS

# Appendix

## A   BACKGROUND

Before discussing the contributions of this work, we provide formal definitions and essential facts that will be repeatedly referenced in the proofs. A summary of the commonly used notation is provided in Table 1.

**Definition A.1** (Proximal operator). The *proximal operator* $\text{prox}_{\gamma f} : \mathbb{R}^d \to \mathbb{R}^d$ of $f$ is defined by

$$\text{prox}_{\gamma f}(x) = \arg\min_{z \in \mathbb{R}^d} \left\{ f(z) + \frac{1}{2\gamma} \|z - x\|^2 \right\},$$

where $\|\cdot\|$ is the standard Euclidean norm.

**Definition A.2** (Moreau envelope). The *Moreau envelope* of an extended-real-valued function $f : \mathbb{R}^d \to \mathbb{R}^d \cup \{\infty\}$ with stepsize $\gamma > 0$ is the function $M_f^\gamma : \mathbb{R}^d \to \mathbb{R}^d$ such that

$$M_f^\gamma(x) = \min_{y \in \mathbb{R}^d} \left\{ f(y) + \frac{1}{2\gamma} \|y - x\|^2 \right\}$$

for all $x \in \mathbb{R}^d$.

### A.1   USEFUL FACTS ABOUT MOREAU ENVELOPE

We will often rely on several useful properties of the Moreau envelopes, summarized below. In what follows, we denote

$$M^\gamma(x) := \frac{1}{n} \sum_{i=1}^n M_{f_i}^\gamma(x).$$

**Lemma A.3** (Beck (2017) Theorem 6.60). *Let $f : \mathbb{R}^d \to \mathbb{R} \cup \{+\infty\}$ be proper, closed and convex. Then its Moreau envelope $M_f^\gamma$ satisfies*

$$\nabla M_f^\gamma(x) = \frac{1}{\gamma}(x - \text{prox}_{\gamma f}(x))$$

*for all $x \in \mathbb{R}^d$, for any $\gamma > 0$.*

**Lemma A.4** ((Li et al., 2024) Lemma 4). *Let $f : \mathbb{R}^d \to \mathbb{R}$ be convex and $L$–smooth. Then $M_f^\gamma$ is $\frac{L}{1+\gamma L}$–smooth.*

**Lemma A.5** ((Li et al., 2024) Lemma 5). *Let $f : \mathbb{R}^d \mapsto \mathbb{R} \cup \{+\infty\}$ be a proper, closed and convex function. Then, for all $\gamma > 0$, $f$ and $M_f^\gamma(x)$ have the same set of minimizers and minimum.*

**Lemma A.6** ((Beck, 2017) Theorem 6.55). *Let $f : \mathbb{R}^d \mapsto \mathbb{R} \cup \{+\infty\}$ be a proper, closed and convex function. Then $M_f^\gamma$ is also a convex function.*

**Lemma A.7** ((Li et al., 2024) Lemma 7). *Let the functions $f_i : \mathbb{R}^d \mapsto \mathbb{R} \cup \{+\infty\}$, $i \in [n]$, be proper, closed, convex and $L_i$–smooth. Then $M^\gamma$ is convex and $L_\gamma$–smooth with*

$$\frac{1}{n^2} \sum_{i=1}^n \frac{L_i}{1 + \gamma L_i} \leq L_\gamma \leq \frac{1}{n} \sum_{i=1}^n \frac{L_i}{1 + \gamma L_i}.$$

**Lemma A.8** ((Li et al., 2024) Lemma 8). *Let the functions $f_i : \mathbb{R}^d \mapsto \mathbb{R} \cup \{+\infty\}$, $i \in [n]$, be proper, closed and convex, and suppose that Assumption 1.3 holds. Then, for all $\gamma \in (0, \infty)$, the function $f(x) = \frac{1}{n} \sum_{i=1}^n f_i(x)$ has the same set of minimizers and minimum as the function $M^\gamma(x)$.*

**Lemma A.9.** *[Li et al. (2024) Lemma 10] Let Assumptions 1.1, 1.2 and 1.3 hold. Then, for any $x_* \in \mathcal{X}_*$ and for all $x \in \mathbb{R}^d$, it holds that*

$$M_{f_i}^\gamma(x) - M_{f_i}^\gamma(x_*) \geq \frac{1}{1 + \gamma L_i} \left( f_i(x) - f_i(x_*) \right).$$

*Consequently,*

$$M^\gamma(x) - M^\gamma(x_*) \geq \frac{1}{1 + \gamma L_{\max}} \left( f(x) - f(x_*) \right).$$

The next lemma will play a role in the analysis of the method when assuming the Polyak-Łojasiewicz (PŁ) condition.

**Assumption A.10** (Polayk-Łojasiewicz condition). *The function $f$ satisfies PŁ condition, i.e., there exists $\mu > 0$ such that*

$$\frac{1}{2} \|\nabla f(x)\|^2 \geq \mu \left(f(x) - f(x_*)\right)$$

*for all $x \in \mathbb{R}^d$.*

**Lemma A.11.** *Let the function $f$ satisfy Assumption A.10 with parameter $\mu^+$, and suppose that Assumptions 1.1, 1.2 and 1.3 hold. Then, Assumption 7.1 holds with parameter*

$$\mu_\gamma^+ \geq \frac{\mu^+}{1 + \gamma L_{\max}}.$$

*Proof.* If $f$ satisfies Assumption A.10, then under Assumption 1.1, Polayk-Łojasiewicz condition and inequality

$$f(x) - f(x_*) \geq \frac{\mu^+}{2} \|x - \Pi(x)\|^2 \quad \forall x \in \mathbb{R}^d, \tag{19}$$

are equivalent (Karimi et al., 2016), where $\Pi(x)$ is the projection of $x$ onto the solution set $\mathcal{X}_*$ of $f$ and $x_* \in \mathcal{X}_*$. Then, using Lemma A.9, we get

$$M^\gamma(x) - M^\gamma(x_*) \geq \frac{1}{1 + \gamma L_{\max}} \left(f(x) - f(x_*)\right) \overset{(19)}{\geq} \frac{\mu^+}{2(1 + \gamma L_{\max})} \|x - \Pi(x)\|^2.$$

Due to Lemma A.8, $M^\gamma$ and $f$ share the set of minimizers. Thus, $\Pi(x)$ is the projection of $x$ onto the solution set $\mathcal{X}_*$ of $M^\gamma$. We can conclude that $M^\gamma$ satisfies (19) (with $M^\gamma$ and $\frac{\mu^+}{1+\gamma L_{\max}}$ instead of $f$ and $\mu^+$), meaning that $M^\gamma$ satisfies PŁ condition with parameter $\frac{\mu^+}{1+\gamma L_{\max}}$. $\qquad\square$

## A.2 NICE SAMPLING

Let us formally introduce the sampling strategy used. Fix a minibatch size $S \in [n]$ and let $\mathcal{S}$ be a random subset of $[n]$ of size $S$, chosen uniformly at random from all $\binom{n}{S}$ subsets of $[n]$ of this size. Such a random set $\mathcal{S}$ is known in the literature under the name *S-nice sampling* (Richtárik & Takáč, 2016).

In the proofs, we will rely on the following useful lemma:

**Lemma A.12.** *Fix $\mathbf{B}_1, \ldots, \mathbf{B}_n \in \mathbb{R}^{k \times l}$ and let $\mathcal{S}$ be an $S$-nice sampling of the indices $[n]$. Then*

$$\mathbb{E}\left[ \left(\frac{1}{S} \sum_{i \in \mathcal{S}} \mathbf{B}_i\right)^\top \left(\frac{1}{S} \sum_{i \in \mathcal{S}} \mathbf{B}_i\right) \right]$$

$$= \frac{n-S}{S(n-1)} \frac{1}{n} \sum_{i=1}^n \mathbf{B}_i^\top \mathbf{B}_i + \frac{n(S-1)}{S(n-1)} \left(\frac{1}{n} \sum_{i=1}^n \mathbf{B}_i\right)^\top \left(\frac{1}{n} \sum_{i=1}^n \mathbf{B}_i\right).$$

*Proof.* Define

$$\chi_i := \begin{cases} 1 & i \in \mathcal{S}, \\ 0 & i \notin \mathcal{S}. \end{cases}$$

Then

$$\mathbb{E}\left[ \left(\frac{1}{S} \sum_{i \in \mathcal{S}} \mathbf{B}_i\right)^\top \left(\frac{1}{S} \sum_{i \in \mathcal{S}} \mathbf{B}_i\right) \right]$$

$$= \mathbb{E}\left[ \left(\frac{1}{S} \sum_{i=1}^n \chi_i \mathbf{B}_i\right)^\top \left(\frac{1}{S} \sum_{i=1}^n \chi_i \mathbf{B}_i\right) \right]$$

$$= \frac{1}{S^2} \mathbb{E} \left[ \sum_{i=1}^{n} \chi_i^2 \mathbf{B}_i^\top \mathbf{B}_i + \sum_{i \neq j} \chi_i \chi_j \mathbf{B}_i^\top \mathbf{B}_j \right]$$

$$= \frac{1}{S^2} \left( \sum_{i=1}^{n} \mathbb{E}\left[\chi_i\right] \mathbf{B}_i^\top \mathbf{B}_i + \sum_{i \neq j} \mathbb{E}\left[\chi_i \chi_j\right] \mathbf{B}_i^\top \mathbf{B}_j \right)$$

$$= \frac{1}{S^2} \left( \frac{S}{n} \sum_{i=1}^{n} \mathbf{B}_i^\top \mathbf{B}_i + \frac{S(S-1)}{n(n-1)} \sum_{i \neq j} \mathbf{B}_i^\top \mathbf{B}_j \right)$$

$$= \frac{1}{S} \left( \frac{1}{n} \sum_{i=1}^{n} \mathbf{B}_i^\top \mathbf{B}_i + \frac{S-1}{n(n-1)} \left( \left( \sum_{i=1}^{n} \mathbf{B}_i \right)^\top \left( \sum_{i=1}^{n} \mathbf{B}_i \right) - \sum_{i=1}^{n} \mathbf{B}_i^\top \mathbf{B}_i \right) \right)$$

$$= \frac{n-S}{S(n-1)} \frac{1}{n} \sum_{i=1}^{n} \mathbf{B}_i^\top \mathbf{B}_i + \frac{n(S-1)}{S(n-1)} \left( \frac{1}{n} \sum_{i=1}^{n} \mathbf{B}_i \right)^\top \left( \frac{1}{n} \sum_{i=1}^{n} \mathbf{B}_i \right).$$

$\square$

### A.3 NOTE ON OTHER SAMPLING STRATEGIES

As discussed in Section 5, Algorithm 2 can handle virtually any (unbiased) sampling technique, with nice sampling used here as an illustrative example. For a comprehensive overview of other potential sampling strategies, we direct the reader to Tyurin et al. (2022), which offers a general framework for analyzing a broad range of sampling mechanisms.

### A.4 USEFUL LEMMAS

**Lemma A.13** ((Karimi et al., 2016) Theorem 2). *Assume that the function $f$ is convex, $L$–smooth, and satisfies the Polyak-Łojasiewicz condition with parameter $\mu$. Then*

$$f(x) - f(x_*) \geq \frac{\mu}{2} \|x - \Pi(x)\|^2,$$

*for all $x \in \mathbb{R}^d$, where $\Pi(x)$ is the projection of $x$ onto the solution set $\mathcal{X}_*$ and $x_* \in \mathcal{X}_*$.*

**Lemma A.14** ((Nesterov, 2018)). *Let $D_f(x, y) := f(x) - f(y) - \langle \nabla f(y), x - y \rangle$ be the Bregman divergence of the function $f$. If $f$ is convex, $L$–smooth and differentiable, then*

$$\frac{1}{L} \|\nabla f(x) - \nabla f(y)\|^2 \leq D_f(x, y) + D_f(y, x)$$

*for any $x, y \in \mathbb{R}^d$.*

---

**Algorithm 1** FedExProx

---

1: **Parameters:** stepsize $\gamma > 0$, extrapolation parameter $\alpha_t > 0$, starting point $x_0 \in \mathbb{R}^d$
2: **for** $k = 0, 1, 2, \dots$ **do**
3: $\quad x_{k+1} = x_k + \alpha_k \left( \frac{1}{n} \sum_{i=1}^n \text{prox}_{\gamma f_i}(x_k) - x_k \right)$
4: **end for**

---

## B  FedExProx AS (S)GD ON MOREAU ENVELOPE REFORMULATION

The key idea behind the original analysis of FedExProx is to rewrite the step

$$x_{k+1} = x_k + \alpha_k \left( \frac{1}{n} \sum_{i=1}^n \text{prox}_{\gamma f_i}(x_k) - x_k \right)$$

in terms of the Moreau envelopes of the local functions. Indeed, if the functions $f_i$ are proper, closed and convex for all $i \in [n]$, by Lemma A.3, the update rule of Algorithm 1 can be rewritten as

$$x_{k+1} = x_k - \alpha_k \gamma \frac{1}{n} \sum_{i=1}^n \nabla M_{f_i}^\gamma(x_k).$$

Analogously, in the partial participation case (Algorithm 2), we have

$$x_{k+1} = x_k - \alpha_k \gamma \frac{1}{S} \sum_{i \in \mathcal{S}_k} \nabla M_{f_i}^\gamma(x_k).$$

It follows that Algorithms 1 and 2 are equivalent to GD and minibatch SGD for minimizing

$$M^\gamma(x) := \frac{1}{n} \sum_{i=1}^n \nabla M_{f_i}^\gamma(x)$$

with stepsize $\alpha_k \gamma$.

## C  FedExProx FOR QUADRATICS

We begin by formally introducing the problem setup and the notation used throughout the proofs.

We are interested in solving the quadratic optimization problem

$$\min_{x \in \mathbb{R}^n} \left\{ f(x) := \frac{1}{n} \sum_{i=1}^{n} \left( \frac{1}{2} x^\top \mathbf{A}_i x - b_i^\top x \right) \right\}, \tag{20}$$

where $\mathbf{A}_i \in \mathrm{Sym}_+^d := \{ \mathbf{X} \in \mathbb{R}^{d \times d} \,|\, \mathbf{X} = \mathbf{X}^\top, \mathbf{X} \succeq 0 \}$ and $b_i \in \mathbb{R}^d$, assuming the interpolation regime (Assumption 1.3), which in this case states that there exists $x_*$ such that $\mathbf{A}_i x_* = b_i$ for all $i \in [n]$.

We denote the set of minimizers by $\mathcal{X}_* := \{ x \in \mathbb{R}^d : \frac{1}{n} \sum_{i=1}^{n} (\mathbf{A}_i x - b_i) = 0 \}$ and let $\Pi(\cdot)$ be the projection onto $\mathcal{X}_*$:

$$\Pi(x) = x - \left( \frac{1}{n} \sum_{i=1}^{n} \mathbf{A}_i \right)^\dagger \left( \frac{1}{n} \sum_{i=1}^{n} (\mathbf{A}_i x - b_i) \right), \tag{21}$$

where $\dagger$ denotes the Moore-Penrose pseudoinverse.

Using eigendecomposition, each matrix $\mathbf{A}_i$, $i \in [n]$ can be decomposed as $\mathbf{A}_i = \mathbf{Q}_i \mathbf{\Lambda}_i \mathbf{Q}_i^\top$, where $\mathbf{Q}_i = [q_{i,1}, \ldots, q_{i,d}] \in \mathbb{R}^{d \times d}$ is the orthogonal matrix whose columns are eigenvectors of $\mathbf{A}_i$ and $\mathbf{\Lambda}_i \in \mathbb{R}^{d \times d}$ is a diagonal matrix whose diagonal elements are the corresponding eigenvalues of $\mathbf{A}_i$, i.e., $\mathbf{\Lambda}_i = \mathrm{diag}(\lambda_1(\mathbf{A}_i), \ldots, \lambda_d(\mathbf{A}_i))$, where $0 = \lambda_1(\mathbf{A}_i) \leq \lambda_2(\mathbf{A}_i) \leq \ldots \leq \lambda_d(\mathbf{A}_i)$. Let $\lambda_{\min}^+(\mathbf{A}_i)$ denote the smallest non-zero eigenvalue of $\mathbf{A}_i$ and $\lambda_{\max}(\mathbf{A}_i)$ – the largest eigenvalue of $\mathbf{A}_i$.

In general, the largest eigenvalue, smallest eigenvalue, and smallest positive eigenvalue of any matrix $\mathbf{B}$ will be denoted by $\lambda_{\max}(\mathbf{B})$, $\lambda_{\min}(\mathbf{B})$ and $\lambda_{\min}^+(\mathbf{B})$, respectively.

The proximal operators in this setting are

$$\mathrm{prox}_{\gamma f_i}(x) = (\gamma \mathbf{A}_i + \mathbf{I})^{-1} (x + \gamma b_i). \tag{22}$$

Thus, by Lemma A.3,

$$\nabla M_{f_i}^\gamma(x) = \frac{1}{\gamma} \left( x - (\gamma \mathbf{A}_i + \mathbf{I})^{-1} (x + \gamma b_i) \right) = \frac{1}{\gamma} \left( \mathbf{I} - (\gamma \mathbf{A}_i + \mathbf{I})^{-1} \right) (x - x_*) \tag{23}$$

for any $x_* \in \mathcal{X}_*$, where we use the fact that $\mathbf{A}_i x_* = b_i$. Furthermore, it can easily be shown that

$$M_{f_i}^\gamma(x) = \frac{1}{2\gamma} \left( x^T (\mathbf{I} - (\gamma \mathbf{A}_i + \mathbf{I})^{-1}) x - 2\gamma b_i^T (\gamma \mathbf{A}_i + \mathbf{I})^{-1} x - \gamma^2 b_i^T (\gamma \mathbf{A}_i + \mathbf{I})^{-1} b_i \right)$$

$$= \frac{1}{2\gamma} \left( (x - x_*)^T (\mathbf{I} - (\gamma \mathbf{A}_i + \mathbf{I})^{-1})(x - x_*) - \gamma x_*^T \mathbf{A}_i x_* \right), \tag{24}$$

and hence

$$M^\gamma(x) = \frac{1}{2} x^T \left( \frac{1}{n} \sum_{i=1}^{n} \frac{1}{\gamma} (\mathbf{I} - (\gamma \mathbf{A}_i + \mathbf{I})^{-1}) \right) x - \left( \frac{1}{n} \sum_{i=1}^{n} (\gamma \mathbf{A}_i + \mathbf{I})^{-1} b_i \right)^T x$$

$$- \frac{\gamma}{2n} \sum_{i=1}^{n} b_i^T (\gamma \mathbf{A}_i + \mathbf{I})^{-1} b_i$$

$$= \frac{1}{2} x^T \mathbf{M} x - \left( \frac{1}{n} \sum_{i=1}^{n} (\gamma \mathbf{A}_i + \mathbf{I})^{-1} b_i \right)^T x - \frac{\gamma}{2n} \sum_{i=1}^{n} b_i^T (\gamma \mathbf{A}_i + \mathbf{I})^{-1} b_i \tag{25}$$

where $\mathbf{M} := \frac{1}{n} \sum_{i=1}^{n} \frac{1}{\gamma} (\mathbf{I} - (\gamma \mathbf{A}_i + \mathbf{I})^{-1})$ is the Hessian matrix of the function $M^\gamma$. This can equivalently be written as

$$M^\gamma(x) = \frac{1}{2} (x - x_*)^T \mathbf{M} (x - x_*) - x_*^T \left( \frac{1}{2n} \sum_{i=1}^{n} \mathbf{A}_i \right) x_* \tag{26}$$

for any $x_* \in \mathcal{X}_*$. We shall denote by $\lambda_{\min}^+(\gamma)$ and $\lambda_{\max}(\gamma)$ the smallest non-zero eigenvalue and the largest eigenvalue of $\mathbf{M}$, respectively.

## C.1 PROPERTIES OF THE HESSIAN

Next, we introduce several important properties of the matrix $\mathbf{M}$, which will be repeatedly used in the proofs that follow.

In the main part of the paper, we stick to the notation $L_\gamma$ and $\mu_\gamma^+$ for easier comparison with the results by Li et al. (2024). However, in the proofs concerning the quadratic case, we adopt a more intuitive eigenvalue-based notation.

**Fact 1.** *Since* $\mathbf{M} = \nabla^2 M^\gamma$*, it holds that* $L_\gamma = \lambda_{\max}(\gamma)$ *and* $\mu_\gamma^+ = \lambda_{\min}^+(\gamma)$*.*

*Proof.* The result is an immediate consequence of (25). $\qquad\square$

**Fact 2.** *Let* $\lambda_{\max}(\gamma)$ *and* $\lambda_{\min}^+(\gamma)$ *be the smallest non-zero eigenvalue and the largest eigenvalue of the matrix*

$$\mathbf{M} = \frac{1}{n}\sum_{i=1}^{n}\frac{1}{\gamma}\left(\mathbf{I} - (\gamma\mathbf{A}_i + \mathbf{I})^{-1}\right),$$

*respectively. Then*

$$\lambda_{\max}(\gamma) = \max_{\|x\|\leq 1}\frac{1}{n}\sum_{i=1}^{n}x^\top\mathbf{Q}_i\left[\frac{\lambda_j(\mathbf{A}_i)}{1+\gamma\lambda_j(\mathbf{A}_i)}\right]_{jj}\mathbf{Q}_i^\top x \qquad (27)$$

*and*

$$\lambda_{\min}^+(\gamma) = \min_{\|x\|=1, x\in\ker(\mathbf{A})^\perp}\frac{1}{n}\sum_{i=1}^{n}x^\top\mathbf{Q}_i\left[\frac{\lambda_j(\mathbf{A}_i)}{1+\gamma\lambda_j(\mathbf{A}_i)}\right]_{jj}\mathbf{Q}_i^\top x, \qquad (28)$$

*where* $\mathbf{A} := \frac{1}{n}\sum_{i=1}^{n}\mathbf{A}_i$ *and* $[b_j]_{jj}$ *is the diagonal matrix with* $b_j$ *as the jth entry.*

*Proof.* Using the eigendecomposition, we can write $\mathbf{M}$ as

$$\mathbf{M} = \frac{1}{n}\sum_{i=1}^{n}\frac{1}{\gamma}\mathbf{Q}_i\left(\mathbf{I} - (\gamma\mathbf{\Lambda}_i + \mathbf{I})^{-1}\right)\mathbf{Q}_i^\top = \frac{1}{n}\sum_{i=1}^{n}\mathbf{Q}_i\left[\frac{\lambda_j(\mathbf{A}_i)}{1+\gamma\lambda_j(\mathbf{A}_i)}\right]_{jj}\mathbf{Q}_i^\top.$$

The result follows from the identities

$$\lambda_{\max}(\gamma) = \max_{\|x\|\leq 1}x^\top\mathbf{M}x$$

and

$$\lambda_{\min}^+(\gamma) = \min_{\|x\|=1, x\in(\ker\mathbf{A})^\perp}x^\top\mathbf{M}x$$

(see Lemma C.7). $\qquad\square$

**Fact 3.** *For any* $\gamma \geq 0$*, it holds that* $\lambda_{\max}(\gamma) \leq \lambda_{\max}(\mathbf{A})$ *and* $\lambda_{\min}^+(\gamma) \leq \lambda_{\min}^+(\mathbf{A})$*.*

*Proof.* From Fact 2, we have

$$\lambda_{\max}(\gamma) \overset{(27)}{=} \max_{\|x\|\leq 1}\frac{1}{n}\sum_{i=1}^{n}x^T\mathbf{Q}_i\left[\frac{\lambda_j(\mathbf{A}_i)}{1+\gamma\lambda_j(\mathbf{A}_i)}\right]_{jj}\mathbf{Q}_i^T x,$$

and similarly

$$\lambda_{\max}(\mathbf{A}) = \max_{\|x\|\leq 1}\frac{1}{n}\sum_{i=1}^{n}x^T\mathbf{Q}_i\left[\lambda_j(\mathbf{A}_i)\right]_{jj}\mathbf{Q}_i^T x.$$

Since it always holds that $\frac{\lambda_j(\mathbf{A}_i)}{1+\gamma\lambda_j(\mathbf{A}_i)} \leq \lambda_j(\mathbf{A}_i)$, we get

$$\lambda_{\max}(\gamma) \leq \lambda_{\max}(\mathbf{A}).$$

The proof of the second inequality is analogous. $\qquad\square$

## C.2  PROOFS FOR SECTION 3

**Theorem 3.2.** *Let Assumptions 1.3 and 3.1 hold. Consider solving a non-strongly convex quadratic optimization problem of the form (1), where $f_i(x) = \frac{1}{2}x^\top \mathbf{A}_i x - b_i^\top x$ for all $i \in [n]$, with $\mathbf{A}_i \in \mathrm{Sym}_+^d$ and $b_i \in \mathbb{R}^d$. Then the assumptions of Theorem 1 by Li et al. (2024) hold, and*

$$T(\gamma) = \pi(\gamma) \times \frac{L_\gamma(1+\gamma L_{\max})R^2}{\varepsilon} \geq \pi(0) \times \frac{LR^2}{\varepsilon} \tag{9}$$

*for all $\gamma > 0$. Moreover, when $\gamma \to 0$, then $\pi(\gamma) \times \frac{L_\gamma(1+\gamma L_{\max})R^2}{\varepsilon} \to \pi(0) \times \frac{LR^2}{\varepsilon}$, and FedExProx effectively reduces to GD.*

*Proof.* We first prove that $L_\gamma(1 + \gamma L_{\max}) \geq L = \lambda_{\max}\left(\frac{1}{n}\sum_{i=1}^n \mathbf{A}_i\right)$ for all $\gamma \geq 0$. Since

$$\nabla^2 M_{f_i}(x) = \frac{1}{\gamma}\left(\mathbf{I} - (\gamma \mathbf{A}_i + \mathbf{I})^{-1}\right),$$

we have

$$L_\gamma = \lambda_{\max}\left(\frac{1}{n}\sum_{i=1}^n \frac{1}{\gamma}\left(\mathbf{I} - (\gamma \mathbf{A}_i + \mathbf{I})^{-1}\right)\right).$$

Therefore, using the fact that $L_{\max} = \max_i \lambda_{\max}(\mathbf{A}_i)$, we obtain

$$(1 + \gamma L_{\max})L_\gamma = \left(1 + \gamma \max_i \lambda_{\max}(\mathbf{A}_i)\right)\frac{1}{\gamma}\lambda_{\max}\left(\frac{1}{n}\sum_{i=1}^n \left(\mathbf{I} - (\gamma \mathbf{A}_i + \mathbf{I})^{-1}\right)\right) \tag{29}$$

$$= \lambda_{\max}\left(\frac{1}{n}\sum_{i=1}^n \frac{1 + \gamma \max_i \lambda_{\max}(\mathbf{A}_i)}{\gamma}\left(\mathbf{I} - (\gamma \mathbf{A}_i + \mathbf{I})^{-1}\right)\right),$$

and hence, it is sufficient to show that

$$\frac{1 + \gamma \max_i \lambda_{\max}(\mathbf{A}_i)}{\gamma}\left(\mathbf{I} - (\gamma \mathbf{A}_i + \mathbf{I})^{-1}\right) \succeq \mathbf{A}_i. \tag{30}$$

Using the eigenvalue decomposition of $\mathbf{A}_i$, the expression can be rewritten as

$$\frac{1 + \gamma \max_i \lambda_{\max}(\mathbf{A}_i)}{\gamma}\left(\mathbf{I} - (\gamma \mathbf{A}_i + \mathbf{I})^{-1}\right) = \frac{1 + \gamma \max_i \lambda_{\max}(\mathbf{A}_i)}{\gamma}\left(\mathbf{I} - (\gamma \mathbf{Q}_i \mathbf{\Lambda}_i \mathbf{Q}_i^\top + \mathbf{I})^{-1}\right)$$

$$= \frac{1 + \gamma \max_i \lambda_{\max}(\mathbf{A}_i)}{\gamma}\mathbf{Q}_i\left(\mathbf{I} - (\gamma \mathbf{\Lambda}_i + \mathbf{I})^{-1}\right)\mathbf{Q}_i^\top.$$

Now, letting $[\mathbf{B}]_j$ be the $j$th diagonal element of matrix $\mathbf{B}$, we have

$$\left[\frac{1 + \gamma \max_i \lambda_{\max}(\mathbf{A}_i)}{\gamma}\left(\mathbf{I} - (\gamma \mathbf{\Lambda}_i + \mathbf{I})^{-1}\right)\right]_j = \frac{1 + \gamma \max_i \lambda_{\max}(\mathbf{A}_i)}{\gamma}\left[\mathbf{I} - (\gamma \mathbf{\Lambda}_i + \mathbf{I})^{-1}\right]_j$$

$$= \frac{1 + \gamma \max_i \lambda_{\max}(\mathbf{A}_i)}{\gamma}\left(1 - \frac{1}{\gamma[\mathbf{\Lambda}_i]_j + 1}\right) = \frac{1 + \gamma \max_i \lambda_{\max}(\mathbf{A}_i)}{\gamma}\frac{\gamma \lambda_j(\mathbf{A}_i)}{1 + \gamma \lambda_j(\mathbf{A}_i)} \geq \lambda_j(\mathbf{A}_i),$$

and hence

$$\frac{(1 + \gamma \max_i \lambda_{\max}(\mathbf{A}_i))}{\gamma}\left(\mathbf{I} - (\gamma \mathbf{\Lambda}_i + \mathbf{I})^{-1}\right) \succeq \mathbf{\Lambda}_i.$$

Multiplying by $\mathbf{Q}_i$ from the left and by $\mathbf{Q}_i^\top$ from the right, we see that (30) holds, and we can conclude that

$$(1 + \gamma L_{\max})L_\gamma \geq \lambda_{\max}\left(\frac{1}{n}\sum_{i=1}^n \mathbf{A}_i\right) = L$$

for all $\gamma \geq 0$. Inequality (9) follows form Assumption 3.1.

We now turn to the second part of the theorem and show that $\pi(\gamma) \times \frac{L_\gamma(1 + \gamma L_{\max})R^2}{\varepsilon} \to \pi(0) \times \frac{LR^2}{\varepsilon}$ as $\gamma \to 0$. The idea behind the proof is that for small enough $\gamma$, it holds that

$$(1 + \gamma L_{\max})L_\gamma \overset{(29)}{=} \left(1 + \gamma \max_i \lambda_{\max}(\mathbf{A}_i)\right) \frac{1}{\gamma} \lambda_{\max}\left(\frac{1}{n}\sum_{i=1}^n \left(\mathbf{I} - (\gamma \mathbf{A}_i + \mathbf{I})^{-1}\right)\right)$$

$$\approx \frac{1}{\gamma}\lambda_{\max}\left(\frac{1}{n}\sum_{i=1}^n \gamma \mathbf{A}_i\right) = \lambda_{\max}\left(\frac{1}{n}\sum_{i=1}^n \mathbf{A}_i\right) = L.$$

More formally, we have

$$\mathbf{I} - (\mathbf{I} + \gamma \mathbf{A}_i)^{-1} = \gamma \mathbf{A}_i - \gamma^2 \mathbf{A}_i^2 + \gamma^3 \mathbf{A}_i^3 + \cdots = \sum_{k=1}^\infty (-1)^{k-1}\gamma^k \mathbf{A}_i^k,$$

so that

$$(1 + \gamma L_{\max})L_\gamma = \left(\frac{1}{\gamma} + \max_i \lambda_{\max}(\mathbf{A}_i)\right)\lambda_{\max}\left(\frac{1}{n}\sum_{i=1}^n \sum_{k=1}^\infty (-1)^{k-1}\gamma^k \mathbf{A}_i^k\right).$$

By applying the dominated convergence theorem to exchange the limit operation and summation, we obtain

$$\lim_{\gamma \to 0}(1 + \gamma L_{\max})L_\gamma = \lim_{\gamma \to 0}\left(\left(1 + \gamma \max_i \lambda_{\max}(\mathbf{A}_i)\right)\lambda_{\max}\left(\frac{1}{n}\sum_{i=1}^n \sum_{k=1}^\infty (-1)^{k-1}\gamma^{k-1} \mathbf{A}_i^k\right)\right)$$

$$= \lambda_{\max}\left(\frac{1}{n}\sum_{i=1}^n \sum_{k=1}^\infty \lim_{\gamma \to 0}(-1)^{k-1}\gamma^{k-1} \mathbf{A}_i^k\right) = \lambda_{\max}\left(\frac{1}{n}\sum_{i=1}^n \mathbf{A}_i\right) = L,$$

as required.

Lastly, letting $\gamma \to 0$, we have $M_{f_i}^\gamma(x) \to f_i(x)$ for all $x$, and consequently $M^\gamma(x) \to f(x)$ (see, e.g., (Rockafellar & Wets, 1998)). Since Algorithm 1 is equivalent to GD for minimizing $M^\gamma(x)$ with stepsize $\alpha\gamma = 1/L_\gamma$, it follows that its iterates converge to those of vanilla GD with stepsize $1/L$. $\qquad\square$

### C.3 TIGHTER ANALYSIS

#### C.3.1 FULL PARTICIPATION CASE

**Iteration complexity.** We first establish the convergence rate of the algorithm.

**Theorem C.1.** *Let Assumption 1.3 hold and let $x_k$ be the iterates of* FedExProx *(Algorithm 1) applied to problem (20). Then*

$$\|x_K - \Pi(x_K)\|^2 \leq \left(1 - \alpha\gamma\left(2 - \alpha\gamma\lambda_{\max}(\gamma)\right)\lambda_{\min}^+(\gamma)\right)^K \|x_0 - \Pi(x_0)\|^2,$$

*where $\lambda_{\max}(\gamma)$ and $\lambda_{\min}^+(\gamma)$ are the largest and the smallest positive eigenvalues of the matrix $\mathbf{M} := \frac{1}{\gamma}\left(\mathbf{I} - \frac{1}{n}\sum_{i=1}^n(\gamma\mathbf{A}_i + \mathbf{I})^{-1}\right)$, respectively.*

*The optimal choice of stepsize $\gamma$ and extrapolation parameter $\alpha$ is $\alpha\gamma = \frac{1}{\lambda_{\max}(\gamma)}$, in which case the rate becomes*

$$\|x_K - \Pi(x_K)\|^2 \leq \left(1 - \frac{\lambda_{\min}^+(\gamma)}{\lambda_{\max}(\gamma)}\right)^K \|x_0 - \Pi(x_0)\|^2.$$

*Hence, the number of iterations needed to reach an $\epsilon$-solution is*

$$K \geq \frac{\lambda_{\max}(\gamma)}{\lambda_{\min}^+(\gamma)} \log\left(\frac{\|x_0 - \Pi(x_0)\|^2}{\epsilon}\right).$$

*Proof of Theorem C.1.* The update rule of FedExProx can be written as

$$x_{k+1} = x_k + \alpha\left(\frac{1}{S}\sum_{i=1}^S \text{prox}_{\gamma f_i}(x_k) - x_k\right)$$

$$\overset{(22)}{=} x_k + \alpha\left(\frac{1}{S}\sum_{i=1}^S (\gamma\mathbf{A}_i + \mathbf{I})^{-1}(x_k + \gamma b_i) - x_k\right)$$

$$= x_k + \alpha\left(\frac{1}{S}\sum_{i=1}^S ((\gamma\mathbf{A}_i + \mathbf{I})^{-1} - \mathbf{I})\right)x_k - \alpha\left(\frac{1}{S}\sum_{i=1}^S ((\gamma\mathbf{A}_i + \mathbf{I})^{-1} - \mathbf{I})\right)\Pi(x_k),$$

where we use the fact that $\mathbf{A}_i x_* = b_i$ for all $x_* \in \mathcal{X}_*$. Therefore, using symmetry of $\mathbf{A}_i$

$$\|x_{k+1} - \Pi(x_{k+1})\|^2 \leq \|x_{k+1} - \Pi(x_k)\|^2$$

$$= \left\|\left((1-\alpha)\mathbf{I} + \alpha\frac{1}{n}\sum_{i=1}^n(\gamma\mathbf{A}_i + \mathbf{I})^{-1}\right)(x_k - \Pi(x_k))\right\|^2$$

$$= \left\|\left(\mathbf{I} - \alpha\underbrace{\left(\frac{1}{n}\sum_{i=1}^n(\mathbf{I} - (\gamma\mathbf{A}_i + \mathbf{I})^{-1})\right)}_{:=\gamma\mathbf{M}}\right)(x_k - \Pi(x_k))\right\|^2$$

$$= (x_k - \Pi(x_k))^T (\mathbf{I} - \alpha\gamma\mathbf{M})^T (\mathbf{I} - \alpha\gamma\mathbf{M})(x_k - \Pi(x_k))$$

$$= (x_k - \Pi(x_k))^T (\mathbf{I} - 2\alpha\gamma\mathbf{M} + \alpha^2\gamma^2\mathbf{M}^2)(x_k - \Pi(x_k)).$$

Since $\mathbf{M}^2 \preceq \lambda_{\max}(\gamma)\mathbf{M}$, it follows that

$$\|x_{k+1} - \Pi(x_{k+1})\|^2 \leq (x_k - \Pi(x_k))^T (\mathbf{I} - 2\alpha\gamma\mathbf{M} + \lambda_{\max}(\gamma)\alpha^2\gamma^2\mathbf{M})(x_k - \Pi(x_k))$$

$$= \|x_k - \Pi(x_k)\|^2 - \alpha\gamma(2 - \lambda_{\max}(\gamma)\alpha\gamma)(x_k - \Pi(x_k))^T\mathbf{M}(x_k - \Pi(x_k))$$

$$\leq \|x_k - \Pi(x_k)\|^2 - \alpha\gamma(2 - \lambda_{\max}(\gamma)\alpha\gamma)\lambda_{\min}^+(\gamma)\|x_k - \Pi(x_k)\|^2$$

for any $\alpha\gamma$ such that $2 - \lambda_{\max}(\gamma)\alpha\gamma \geq 0$, where the last inequality follows from the fact that $x_k - \Pi(x_k) \in \text{range}(\mathbf{M})$ (see Lemma C.6). Consequently,

$$\|x_{k+1} - \Pi(x_{k+1})\|^2 \leq \left(1 - \alpha\gamma(2 - \alpha\gamma\lambda_{\max}(\gamma))\lambda_{\min}^+(\gamma)\right)\|x_k - \Pi(x_k)\|^2.$$

It remains to unroll the recurrence and substitute the optimal step size $\alpha\gamma = \frac{1}{\lambda_{\max}(\gamma)}$ to obtain the rate

$$\|x_{k+1} - \Pi(x_{k+1})\|^2 \leq \left(1 - \frac{\lambda_{\min}^+(\gamma)}{\lambda_{\max}(\gamma)}\right)\|x_k - \Pi(x_k)\|^2.$$

□

**Theorem 4.1.** *Fix any $\gamma > 0$ and consider solving non-strongly convex quadratic optimization problem* (1) *where $f_i(x) = \frac{1}{2}x^\top \mathbf{A}_i x - b_i^\top x$ for all $i \in [n]$, with $\mathbf{A}_i \in \mathrm{Sym}_+^d$ and $b_i \in \mathbb{R}^d$. Under Assumption 1.3, FedExProx with $\alpha = \frac{1}{\gamma L_\gamma}$ finds $\bar{x}$ such that $\mathbb{E}[f(\bar{x})] - f(x_*) \leq \varepsilon$ after*

$$\mathcal{O}\left(\frac{L_\gamma}{\mu_\gamma^+}\log\frac{1}{\varepsilon}\right) \tag{10}$$

*iterations, where $L_\gamma$ is a smoothness constant of $M^\gamma$ and $\mu_\gamma^+$ is the smallest non-zero eigenvalue of the matrix $\nabla^2 M^\gamma$.*

*Proof.* The result follows directly from Theorem C.1, smoothness of $f$, and Fact 1 by substituting $\lambda_{\max}(\gamma) = \lambda_{\max}(\nabla^2 M^\gamma) = L_\gamma$ and $\lambda_{\min}^+(\gamma) = \lambda_{\min}^+(\nabla^2 M^\gamma) = \mu_\gamma^+$. □

*Remark* C.2. Recall from the proof of Fact 2 that

$$\mathbf{M} = \frac{1}{n}\sum_{i=1}^n \mathbf{Q}_i \left[\frac{\lambda_j(\mathbf{A}_i)}{1+\gamma\lambda_j(\mathbf{A}_i)}\right]_{jj}\mathbf{Q}_i^\top,$$

where $\frac{\lambda_j(\mathbf{A}_i)}{1+\gamma\lambda_j(\mathbf{A}_i)} \in \left[\frac{\lambda_j(\mathbf{A}_i)}{2}, \lambda_j(\mathbf{A}_i)\right]$ if $\gamma \leq \frac{1}{\lambda_j(\mathbf{A}_i)}$. Consequently, for $\gamma \leq \frac{1}{\max_{i\in[n]}\lambda_{\max}(\mathbf{A}_i)}$, we have

$$\mathbf{M} = \frac{1}{n}\sum_{i=1}^n \mathbf{Q}_i \left[\frac{\lambda_j(\mathbf{A}_i)}{1+\gamma\lambda_j(\mathbf{A}_i)}\right]_{jj}\mathbf{Q}_i^\top \preceq \frac{1}{n}\sum_{i=1}^n \mathbf{Q}_i \left[\lambda_j(\mathbf{A}_i)\right]_{jj}\mathbf{Q}_i^\top = \frac{1}{n}\sum_{i=1}^n \mathbf{A}_i = \mathbf{A},$$

and analogously, $\mathbf{M} \succeq \frac{1}{2}\mathbf{A}$. Using (27) and (28) gives

$$\frac{1}{2}\lambda_{\max}(\mathbf{A}) = \frac{1}{2}\max_{\|x\|\leq 1} x^\top \mathbf{A}x \leq \lambda_{\max}(\gamma) = \max_{\|x\|\leq 1} x^\top \mathbf{M}x \leq \max_{\|x\|\leq 1} x^\top \mathbf{A}x = \lambda_{\max}(\mathbf{A}),$$

and similarly

$$\frac{1}{2}\lambda_{\min}^+(\mathbf{A}) \leq \lambda_{\min}^+(\gamma) = \min_{\|x\|=1, x\in(\ker\mathbf{A})^\perp} x^\top \mathbf{M}x \leq \lambda_{\min}^+(\mathbf{A}).$$

Noting that $L_\gamma = \lambda_{\max}(\gamma)$, $\mu_\gamma^+ = \lambda_{\min}^+(\gamma)$, $L = \lambda_{\max}(\mathbf{A})$ and $\mu^+ = \lambda_{\min}^+(\mathbf{A})$, it follows that the rates (10) of FedExProx and (32) of GD coincide up to a constant factor when $\gamma$ is sufficiently small. This should come as no surprise, as Theorem 3.2 shows that FedExProx effectively reduces to GD as $\gamma \to 0$.

**Time complexity.** Let us now prove the result from Section 4.1. We assume that each worker $i$ computes the proximal operator using an iterative method, with the running time proportional to the condition number of the subproblem. Given that $\lambda_{\min}(\mathbf{A}_i) = 0$ for all $i \in [n]$ (otherwise the problem becomes trivial), the time required for all clients to compute $\mathrm{prox}_{\gamma f_i}(x_k)$ at each global iteration $k$ is

$$\tau \times \left(1 + \gamma\max_{i\in[n]}\lambda_{\max}(\mathbf{A}_i)\right)$$

seconds, where $\tau$ is the time per one iteration of solving the subproblem. Then, it takes $\mu$ seconds to aggregate the updates at the server and move on to the next step. Since, according to Theorem C.1, the number of iterations needed to reach an $\epsilon$-solution is

$$K = \frac{\lambda_{\max}(\gamma)}{\lambda_{\min}^+(\gamma)}\log\left(\frac{\|x_0 - \Pi(x_0)\|^2}{\epsilon}\right),$$

the total time required to solve the global problem is

$$T_\mu(\gamma) := \frac{\lambda_{\max}(\gamma)}{\lambda_{\min}^+(\gamma)}\log\left(\frac{\|x_0 - \Pi(x_0)\|^2}{\epsilon}\right) \times \left(\mu + \tau \times \left(1 + \gamma\max_{i\in[n]}\lambda_{\max}(\mathbf{A}_i)\right)\right). \tag{31}$$

**Theorem 4.3.** *Consider the non-strongly convex quadratic optimization problem from Theorem 4.1. Up to a constant factor, the time complexity (12) is minimized by $\gamma \in \left[\frac{1}{\max_{i \in [n]} \lambda_{\max}(\mathbf{A}_i)}, \min\left\{\frac{\frac{\mu}{\tau}-1}{\max_{i \in [n]} \lambda_{\max}(\mathbf{A}_i)}, \frac{1}{\min_{i \in [n]} \lambda_{\min}^+(\mathbf{A}_i)}\right\}\right]$ if $\frac{\mu}{\tau} \geq 2$ and by $\gamma \in \left[0, \max\left\{0, \min\left\{\frac{\frac{\mu}{\tau}-1}{\max_{i \in [n]} \lambda_{\max}(\mathbf{A}_i)}, \frac{1}{\min_{i \in [n]} \lambda_{\min}^+(\mathbf{A}_i)}\right\}\right\}\right]$ if $\frac{\mu}{\tau} < 2$, and $T_\mu(\gamma) \leq T_{\mathsf{GD}}$.*

*Proof.* Recall that $\lambda_{\max}(\gamma)$ and $\lambda_{\min}^+(\gamma)$ are the eigenvalues of the matrix

$$\mathbf{M} = \frac{1}{n} \sum_{i=1}^{n} \frac{1}{\gamma} \left(\mathbf{I} - (\gamma \mathbf{A}_i + \mathbf{I})^{-1}\right) \overset{(2)}{=} \frac{1}{n} \sum_{i=1}^{n} \mathbf{Q}_i \left[\frac{\lambda_j(\mathbf{A}_i)}{1 + \gamma \lambda_j(\mathbf{A}_i)}\right]_{jj} \mathbf{Q}_i^\top.$$

Since if $\gamma \geq \frac{1}{\lambda_j(\mathbf{A}_i)}$ and $\lambda_j(\mathbf{A}_i) > 0$, then $\frac{\gamma \lambda_j(\mathbf{A}_i)}{1+\gamma \lambda_j(\mathbf{A}_i)} \in \left[\frac{1}{2}, 1\right)$, and if $\gamma \leq \frac{1}{\lambda_j(\mathbf{A}_i)}$, then $\frac{\gamma \lambda_j(\mathbf{A}_i)}{1+\gamma \lambda_j(\mathbf{A}_i)} \in \left[\frac{\gamma \lambda_j(\mathbf{A}_i)}{2}, \gamma \lambda_j(\mathbf{A}_i)\right)$, it follows that

$$\frac{\lambda_j(\mathbf{A}_i)}{1 + \gamma \lambda_j(\mathbf{A}_i)} = \begin{cases} \Theta\left(\frac{1}{\gamma}\right), & \gamma \geq \frac{1}{\lambda_j(\mathbf{A}_i)}, \\ \Theta\left(\lambda_j(\mathbf{A}_i)\right), & \gamma < \frac{1}{\lambda_j(\mathbf{A}_i)}. \end{cases} \tag{32}$$

Moreover, the identities (27) and (28) from Fact 2 tell us that

$$\lambda_{\max}(\gamma) = \max_{\|x\| \leq 1} \frac{1}{n} \sum_{i=1}^{n} x^\top \mathbf{Q}_i \left[\frac{\lambda_j(\mathbf{A}_i)}{1 + \gamma \lambda_j(\mathbf{A}_i)}\right]_{jj} \mathbf{Q}_i^\top x$$

and

$$\lambda_{\min}^+(\gamma) = \min_{\|x\|=1, x \in \ker(\mathbf{A})^\perp} \frac{1}{n} \sum_{i=1}^{n} x^\top \mathbf{Q}_i \left[\frac{\lambda_j(\mathbf{A}_i)}{1 + \gamma \lambda_j(\mathbf{A}_i)}\right]_{jj} \mathbf{Q}_i^\top x,$$

where $\mathbf{A} := \frac{1}{n} \sum_{i=1}^{n} \mathbf{A}_i$. Due to (27), (28) and (32), the function $T_\mu(\gamma)$ is approximately constant for all $\gamma \leq \frac{1}{\max_{i \in [n]} \lambda_{\max}(\mathbf{A}_i)}$. If $\gamma \geq \frac{\frac{\mu}{\tau}-1}{\max_{i \in [n]} \lambda_{\max}(\mathbf{A}_i)}$ then $\tau(1 + \gamma \max_{i \in [n]} \lambda_{\max}(\mathbf{A}_i)) \geq \mu$, and consequently,

$$T_\mu(\gamma) = \frac{\lambda_{\max}(\gamma)}{\lambda_{\min}^+(\gamma)} \log\left(\frac{\|x_0 - \Pi(x_0)\|^2}{\epsilon}\right) \times \left(\mu + \tau\left(1 + \gamma \max_{i \in [n]} \lambda_{\max}(\mathbf{A}_i)\right)\right)$$

$$\leq 2\tau \frac{\lambda_{\max}(\gamma)}{\lambda_{\min}^+(\gamma)} \log\left(\frac{\|x_0 - \Pi(x_0)\|^2}{\epsilon}\right) \times \left(1 + \gamma \max_{i \in [n]} \lambda_{\max}(\mathbf{A}_i)\right) := 2\bar{g}(\gamma).$$

Hence, $\bar{g}(\gamma) \leq T_\mu(\gamma) \leq 2\bar{g}(\gamma)$, and the term to be minimized is

$$\bar{T}_\mu(\gamma) := \frac{\lambda_{\max}(\gamma)}{\lambda_{\min}^+(\gamma)} \times \left(1 + \gamma \max_{i \in [n]} \lambda_{\max}(\mathbf{A}_i)\right).$$

Using (27) and (28), this can be written as

$$\bar{T}_\mu(\gamma) = \frac{\max_{\|x\| \leq 1} \frac{1}{n} \sum_{i=1}^{n} x^\top \mathbf{Q}_i \left[\frac{1 + \gamma \max_{i \in [n]} \lambda_{\max}(\mathbf{A}_i)}{1 + \gamma \lambda_j(\mathbf{A}_i)} \lambda_j(\mathbf{A}_i)\right]_{jj} \mathbf{Q}_i^\top x}{\min_{\|x\|=1, x \in \ker(\mathbf{A})^\perp} \frac{1}{n} \sum_{i=1}^{n} x^\top \mathbf{Q}_i \left[\frac{\lambda_j(\mathbf{A}_i)}{1 + \gamma \lambda_j(\mathbf{A}_i)}\right]_{jj} \mathbf{Q}_i^\top x},$$

where the numerator is non-decreasing and the denominator is non-increasing as a function of $\gamma$. It follows that $\bar{g}(\gamma)$ is non-decreasing for $\gamma \geq \frac{\frac{\mu}{\tau}-1}{\max_{i \in [n]} \lambda_{\max}(\mathbf{A}_i)}$. Hence, when $\frac{\mu}{\tau} \geq 2$, the optimal (up to a multiplicative factor) $\gamma$ belongs to the interval

$$\left[\frac{1}{\max_{i \in [n]} \lambda_{\max}(\mathbf{A}_i)}, \frac{\frac{\mu}{\tau}-1}{\max_{i \in [n]} \lambda_{\max}(\mathbf{A}_i)}\right]. \tag{33}$$

When $\frac{\mu}{\tau} < 2$, the time-complexity is non-decreasing for $\gamma \geq \frac{\frac{\mu}{\tau}-1}{\max_{i \in [n]} \lambda_{\max}(\mathbf{A}_i)}$ and approximately constant otherwise, so the choice

$$\gamma \in \left[0, \max\left\{0, \frac{\frac{\mu}{\tau}-1}{\max_{i \in [n]} \lambda_{\max}(\mathbf{A}_i)}\right\}\right] \tag{34}$$

---

**Algorithm 2** FedExProx with partial participation

---

1: **Parameters:** stepsize $\gamma > 0$, extrapolation parameter $\alpha_k > 0$, starting point $x_0 \in \mathbb{R}^d$, batch size $S$
2: **for** $k = 0, 1, 2, \ldots$ **do**
3:     Sample a minibatch $\mathcal{S}_k \subseteq [n]$ uniformly from all subsets of cardinality $S$
4:     $x_{k+1} = x_k + \alpha_k \left( \frac{1}{S} \sum_{i \in \mathcal{S}_k} \text{prox}_{\gamma f_i}(x_k) - x_k \right)$
5: **end for**

---

is optimal (again, up to a constant factor).

Now, suppose that $\gamma \geq \frac{1}{\min_{i \in [n]} \lambda_{\min}^+(\mathbf{A}_i)}$. In this case, the diagonal entries of the matrix $\left[ \frac{\gamma \lambda_j(\mathbf{A}_i)}{1 + \gamma \lambda_j(\mathbf{A}_i)} \right]_{jj}$ are either to 0 (when $\lambda_j(\mathbf{A}_i) = 0$) or lie within the interval $\left[ \frac{1}{2}, 1 \right)$. Therefore, the ratio

$$\frac{\lambda_{\max}(\gamma)}{\lambda_{\min}^+(\gamma)} = \frac{\max_{\|x\| \leq 1} \frac{1}{n} \sum_{i=1}^n x^\top \mathbf{Q}_i \left[ \frac{\gamma \lambda_j(\mathbf{A}_i)}{1 + \gamma \lambda_j(\mathbf{A}_i)} \right]_{jj} \mathbf{Q}_i^\top x}{\min_{\|x\|=1, x \in \ker(\mathbf{A})^\perp} \frac{1}{n} \sum_{i=1}^n x^\top \mathbf{Q}_i \left[ \frac{\gamma \lambda_j(\mathbf{A}_i)}{1 + \gamma \lambda_j(\mathbf{A}_i)} \right]_{jj} \mathbf{Q}_i^\top x}$$

is approximately constant in $\gamma$. Consequently, the time

$$T_\mu(\gamma) := \frac{\lambda_{\max}(\gamma)}{\lambda_{\min}^+(\gamma)} \log \left( \frac{\|x_0 - \Pi(x_0)\|^2}{\epsilon} \right) \times \left( \mu + \tau \left( 1 + \gamma \max_{i \in [n]} \lambda_{\max}(\mathbf{A}_i) \right) \right)$$

is an increasing function of $\gamma$ for $\gamma \geq \frac{1}{\min_{i \in [n]} \lambda_{\min}^+(\mathbf{A}_i)}$, and the optimal range for $\gamma$ can be tightened from the previously established bounds (33) and (34) to

$$\left[ \frac{1}{\max_{i \in [n]} \lambda_{\max}(\mathbf{A}_i)}, \min \left\{ \frac{\frac{\mu}{\tau} - 1}{\max_{i \in [n]} \lambda_{\max}(\mathbf{A}_i)}, \frac{1}{\min_{i \in [n]} \lambda_{\min}^+(\mathbf{A}_i)} \right\} \right]$$

when $\frac{\mu}{\tau} \geq 2$ and

$$\gamma \in \left[ 0, \max \left\{ 0, \min \left\{ \frac{\frac{\mu}{\tau} - 1}{\max_{i \in [n]} \lambda_{\max}(\mathbf{A}_i)}, \frac{1}{\min_{i \in [n]} \lambda_{\min}^+(\mathbf{A}_i)} \right\} \right\} \right]$$

when $\frac{\mu}{\tau} < 2$.

$\square$

### C.3.2    PARTIAL PARTICIPATION CASE

**Iteration complexity.** We again start with establishing the iteration complexity of the algorithm.

**Theorem C.3.** *Let Assumption 1.3 hold and let $x_k$ be the iterates of minibatch* FedExProx *(Algorithm 2) applied to problem (20). Then*

$$\mathbb{E} \left[ \|x_K - \Pi(x_K)\|^2 \right] \leq \left( 1 - \alpha \gamma \left( 2 - \alpha \gamma L_{\gamma, S} \right) \lambda_{\min}^+(\gamma) \right)^K \|x_0 - \Pi(x_0)\|^2,$$

*where $\lambda_{\min}^+(\gamma)$ is the smallest positive eigenvalue of the matrix $\mathbf{M} := \frac{1}{\gamma} \left( \mathbf{I} - \frac{1}{n} \sum_{i=1}^n (\gamma \mathbf{A}_i + \mathbf{I})^{-1} \right)$ and $L_{\gamma, S} := \frac{n - S}{S(n-1)} \frac{\max_{i \in [n]} \lambda_{\max}(\mathbf{A}_i)}{1 + \gamma \max_{i \in [n]} \lambda_{\max}(\mathbf{A}_i)} + \frac{n(S-1)}{S(n-1)} \lambda_{\max}(\gamma)$.*

*The optimal choice of stepsize $\gamma$ and extrapolation parameter $\alpha$ is $\alpha \gamma = \frac{1}{L_{\gamma, S}}$, in which case the rate becomes*

$$\mathbb{E} \left[ \|x_K - \Pi(x_K)\|^2 \right] \leq \left( 1 - \frac{\lambda_{\min}^+(\gamma)}{L_{\gamma, S}} \right)^K \|x_0 - \Pi(x_0)\|^2.$$

*Hence, the number of iterations needed to reach an $\epsilon$-solution is*

$$K \geq \frac{L_{\gamma, S}}{\lambda_{\min}^+(\gamma)} \log \left( \frac{\|x_0 - \Pi(x_0)\|^2}{\epsilon} \right). \tag{35}$$

*Remark* C.4. Similar to the proof of Theorem 4.1, the result of Theorem 5.1 follows directly from Theorem C.3, smoothness of $f$, and Fact 1 by noting that $\lambda_{\max}(\gamma) = \lambda_{\max}(\nabla^2 M^\gamma) = L_\gamma$ and $\lambda_{\min}^+(\gamma) = \lambda_{\min}^+(\nabla^2 M^\gamma) = \mu_\gamma^+$.

*Proof of Theorem C.3.* The update rule of FedExProx can be written as

$$x_{k+1} = x_k + \alpha \left( \frac{1}{S} \sum_{i \in \mathcal{S}_k} \mathrm{prox}_{\gamma f_i}(x_k) - x_k \right)$$

$$= x_k + \alpha \left( \frac{1}{S} \sum_{i \in \mathcal{S}_k} (\gamma \mathbf{A}_i + \mathbf{I})^{-1}(x_k + \gamma b_i) - x_k \right)$$

$$= x_k + \alpha \left( \frac{1}{S} \sum_{i \in \mathcal{S}_k} ((\gamma \mathbf{A}_i + \mathbf{I})^{-1} - \mathbf{I}) \right) x_k - \alpha \left( \frac{1}{S} \sum_{i \in \mathcal{S}_k} ((\gamma \mathbf{A}_i + \mathbf{I})^{-1} - \mathbf{I}) \right) \Pi(x_k),$$

where we use the fact that $\mathbf{A}_i x_* = b_i$ for all $x_* \in \mathcal{X}_*$. Therefore

$$\|x_{k+1} - \Pi(x_{k+1})\|^2$$
$$\leq \|x_{k+1} - \Pi(x_k)\|^2$$
$$= \left\| \left( (1-\alpha)\mathbf{I} + \alpha \frac{1}{S} \sum_{i \in \mathcal{S}_k} (\gamma \mathbf{A}_i + \mathbf{I})^{-1} \right) (x_k - \Pi(x_k)) \right\|^2$$
$$= \|x_k - \Pi(x_k)\|^2 - 2\alpha(x_k - \Pi(x_k))^T \left( \mathbf{I} - \frac{1}{S} \sum_{i \in \mathcal{S}_k} (\gamma \mathbf{A}_i + \mathbf{I})^{-1} \right) (x_k - \Pi(x_k))$$
$$+ \alpha^2 (x_k - \Pi(x_k))^T \left( \mathbf{I} - \frac{1}{S} \sum_{i \in \mathcal{S}_k} (\gamma \mathbf{A}_i + \mathbf{I})^{-1} \right)^T \left( \mathbf{I} - \frac{1}{S} \sum_{i \in \mathcal{S}_k} (\gamma \mathbf{A}_i + \mathbf{I})^{-1} \right) (x_k - \Pi(x_k))$$
$$= \|x_k - \Pi(x_k)\|^2 - 2\alpha\gamma(x_k - \Pi(x_k))^T \mathbf{M}_k (x_k - \Pi(x_k))$$
$$+ \alpha^2 \gamma^2 (x_k - \Pi(x_k))^T \mathbf{M}_k^T \mathbf{M}_k (x_k - \Pi(x_k)), \tag{36}$$

where $\mathbf{M}_k := \frac{1}{\gamma} \left( \mathbf{I} - \frac{1}{S} \sum_{i \in \mathcal{S}_k} (\gamma \mathbf{A}_i + \mathbf{I})^{-1} \right)$. Next, using the fact that $\mathbf{M}^2 \preceq \lambda_{\max}(\gamma)\mathbf{M}$ and applying Lemma A.12 with $\mathbf{B}_i = \frac{1}{\gamma} \left( \mathbf{I} - (\gamma \mathbf{A}_i + \mathbf{I})^{-1} \right)$, the expectation of $\mathbf{M}_k^T \mathbf{M}_k$ is

$$\mathbb{E}\left[\mathbf{M}_k^T \mathbf{M}_k\right] = \frac{1}{\gamma^2} \frac{n-S}{S(n-1)} \frac{1}{n} \sum_{i=1}^n \left( \mathbf{I} - (\gamma \mathbf{A}_i + \mathbf{I})^{-1} \right)^\top \left( \mathbf{I} - (\gamma \mathbf{A}_i + \mathbf{I})^{-1} \right) + \frac{n(S-1)}{S(n-1)} \mathbf{M}^\top \mathbf{M}$$

$$= \frac{1}{\gamma^2} \frac{n-S}{S(n-1)} \frac{1}{n} \sum_{i=1}^n \left( \mathbf{I} - (\gamma \mathbf{A}_i + \mathbf{I})^{-1} \right)^2 + \frac{n(S-1)}{S(n-1)} \mathbf{M}^2$$

$$\leq \frac{1}{\gamma} \frac{n-S}{S(n-1)} \max_{i \in [n]} \lambda_{\max} \left( \mathbf{I} - (\gamma \mathbf{A}_i + \mathbf{I})^{-1} \right) \frac{1}{n} \sum_{i=1}^n \frac{1}{\gamma} \left( \mathbf{I} - (\gamma \mathbf{A}_i + \mathbf{I})^{-1} \right)$$

$$+ \frac{n(S-1)}{S(n-1)} \lambda_{\max}(\gamma)\mathbf{M}$$

$$= \frac{1}{\gamma} \frac{n-S}{S(n-1)} \max_{i \in [n]} \left( \frac{\gamma \lambda_{\max}(\mathbf{A}_i)}{1 + \gamma \lambda_{\max}(\mathbf{A}_i)} \right) \mathbf{M} + \frac{n(S-1)}{S(n-1)} \lambda_{\max}(\gamma)\mathbf{M}$$

$$= \underbrace{\left( \frac{n-S}{S(n-1)} \frac{\max_{i \in [n]} \lambda_{\max}(\mathbf{A}_i)}{1 + \gamma \max_{i \in [n]} \lambda_{\max}(\mathbf{A}_i)} + \frac{n(S-1)}{S(n-1)} \lambda_{\max}(\gamma) \right)}_{:=L_{\gamma,S}} \mathbf{M}. \tag{37}$$

Hence, taking expectation conditioned on $x_k$ in (36) gives

$$\mathbb{E}_k \left[ \|x_{k+1} - \Pi(x_{k+1})\|^2 \right]$$

$$\leq \|x_k - \Pi(x_k)\|^2 + \alpha^2\gamma^2(x_k - \Pi(x_k))^T\mathbb{E}_k\left[\mathbf{M}_k^T\mathbf{M}_k\right](x_k - \Pi(x_k))$$
$$-2\alpha\gamma(x_k - \Pi(x_k))^T\mathbb{E}_k\left[\mathbf{M}_k\right](x_k - \Pi(x_k))$$
$$\overset{(37)}{=} \|x_k - \Pi(x_k)\|^2 + \alpha^2\gamma^2 L_{\gamma,S}(x_k - \Pi(x_k))^T\mathbf{M}(x_k - \Pi(x_k))$$
$$-2\alpha\gamma(x_k - \Pi(x_k))^T\mathbf{M}(x_k - \Pi(x_k))$$
$$= \|x_k - \Pi(x_k)\|^2 - \alpha\gamma\left(2 - \alpha\gamma L_{\gamma,S}\right)(x_k - \Pi(x_k))^T\mathbf{M}(x_k - \Pi(x_k))$$
$$\leq \|x_k - \Pi(x_k)\|^2 - \alpha\gamma\left(2 - \alpha\gamma L_{\gamma,S}\right)\lambda^+_{\min}(\gamma)(x_k - \Pi(x_k))^T(x_k - \Pi(x_k)),$$

where the last inequality follows from the fact that $x_k - \Pi(x_k) \in \text{range}(\mathbf{M})$ (see Lemma C.6). Taking expectation again,

$$\mathbb{E}\left[\|x_{k+1} - \Pi(x_{k+1})\|^2\right] \leq \left(1 - \alpha\gamma\left(2 - \alpha\gamma L_{\gamma,S}\right)\lambda^+_{\min}(\gamma)\right)\mathbb{E}\left[\|x_k - \Pi(x_k)\|^2\right].$$

It remains to unroll the recurrence. $\qquad\square$

**Time complexity.** Let us consider the same setup as in Section C.3.1, i.e., at each iteration $k$, each client computes a proximal operator $\text{prox}_{\gamma f_i}(x_k)$ in at most

$$\tau \times \left(1 + \gamma\max_{i\in[n]}\lambda_{\max}(\mathbf{A}_i)\right)$$

seconds, where $\tau$ is the time per one iteration of solving the subproblem, and it takes $\mu$ seconds to aggregate the updates at the server and move on to the next step. According to Theorem C.3, the number of iterations needed to reach an $\epsilon$-solution is

$$K = \frac{L_{\gamma,S}}{\lambda^+_{\min}(\gamma)}\log\left(\frac{\|x_0 - \Pi(x_0)\|^2}{\epsilon}\right),$$

where $L_{\gamma,S} := \frac{n-S}{S(n-1)}\frac{\max_{i\in[n]}\lambda_{\max}(\mathbf{A}_i)}{1+\gamma\max_{i\in[n]}\lambda_{\max}(\mathbf{A}_i)} + \frac{n(S-1)}{S(n-1)}\lambda_{\max}(\gamma)$. Hence, the total time required to solve the global problem is at most

$$T_\mu(\gamma, S) := \frac{L_{\gamma,S}}{\lambda^+_{\min}(\gamma)}\log\left(\frac{\|x_0 - \Pi(x_0)\|^2}{\epsilon}\right) \times \left(\mu + \tau \times \left(1 + \gamma\max_{i\in[n]}\lambda_{\max}(\mathbf{A}_i)\right)\right). \quad (38)$$

**Theorem 5.2.** *Up to a constant factor, the time complexity (14) is minimized by* $\gamma \in \left[\frac{1}{\max_{i\in[n]}\lambda_{\max}(\mathbf{A}_i)}, \min\left\{\frac{\frac{\mu}{\tau}-1}{\max_{i\in[n]}\lambda_{\max}(\mathbf{A}_i)}, \frac{1}{\min_{i\in[n]}\lambda^+_{\min}(\mathbf{A}_i)}\right\}\right]$ *if* $\frac{\mu}{\tau} \geq 2$ *and by* $\gamma \in \left[0, \max\left\{0, \min\left\{\frac{\frac{\mu}{\tau}-1}{\max_{i\in[n]}\lambda_{\max}(\mathbf{A}_i)}, \frac{1}{\min_{i\in[n]}\lambda^+_{\min}(\mathbf{A}_i)}\right\}\right\}\right]$ *if* $\frac{\mu}{\tau} < 2$.

*Proof.* Fix some $S \in [n]$. Then, we are interested in minimizing

$$\bar{T}_\mu(\gamma) = \frac{n-S}{S(n-1)}\underbrace{\frac{1}{\lambda^+_{\min}(\gamma)}\frac{\max_{i\in[n]}\lambda_{\max}(\mathbf{A}_i)}{1+\gamma\max_{i\in[n]}\lambda_{\max}(\mathbf{A}_i)}\left(\mu + \tau\left(1 + \gamma\max_{i\in[n]}\lambda_{\max}(\mathbf{A}_i)\right)\right)}_{:=\bar{g}_1(\gamma)}$$

$$+ \frac{n(S-1)}{S(n-1)}\underbrace{\frac{\lambda_{\max}(\gamma)}{\lambda^+_{\min}(\gamma)}\left(\mu + \tau\left(1 + \gamma\max_{i\in[n]}\lambda_{\max}(\mathbf{A}_i)\right)\right)}_{:=\bar{g}_2(\gamma)}$$

$$:= \frac{n-S}{S(n-1)}\bar{g}_1(\gamma) + \frac{n(S-1)}{S(n-1)}\bar{g}_2(\gamma).$$

We know from Theorem 4.3 that the second term of this expression, $\bar{g}_2(\gamma)$, is minimized by

$$\left[\frac{1}{\max_{i\in[n]}\lambda_{\max}(\mathbf{A}_i)}, \min\left\{\frac{\frac{\mu}{\tau}-1}{\max_{i\in[n]}\lambda_{\max}(\mathbf{A}_i)}, \frac{1}{\min_{i\in[n]}\lambda^+_{\min}(\mathbf{A}_i)}\right\}\right]$$

when $\frac{\mu}{\tau} \geq 2$ and

$$\gamma \in \left[0, \max\left\{0, \min\left\{\frac{\frac{\mu}{\tau} - 1}{\max_{i \in [n]} \lambda_{\max}(\mathbf{A}_i)}, \frac{1}{\min_{i \in [n]} \lambda_{\min}^+(\mathbf{A}_i)}\right\}\right\}\right]$$

when $\frac{\mu}{\tau} < 2$.

The same argument can be applied to the term $\bar{g}_1(\gamma)$. To be more precise, due to (27), (28), and (32), the function $\bar{g}_1(\gamma)$ is approximately constant for all $\gamma \leq \frac{1}{\max_{i \in [n]} \lambda_{\max}(\mathbf{A}_i)}$. For $\gamma > \frac{\frac{\mu}{\tau} - 1}{\max_{i \in [n]} \lambda_{\max}(\mathbf{A}_i)}$, we have $\frac{\mu}{\tau} + 1 + \gamma \max_{i \in [n]} \lambda_{\max}(\mathbf{A}_i) < 2(1 + \gamma \max_{i \in [n]} \lambda_{\max}(\mathbf{A}_i))$. Consequently, letting $\bar{g}(\gamma) := 2\tau \frac{\max_{i \in [n]} \lambda_{\max}(\mathbf{A}_i)}{\lambda_{\min}^+(\gamma)}$, $\bar{g}_1(\gamma)$ can be bounded as

$$\frac{1}{2}\bar{g}(\gamma) < \bar{g}_1(\gamma) < \bar{g}(\gamma).$$

Since $\lambda_{\min}^+(\gamma)$ is a non-increasing function of $\gamma$, $\bar{g}(\gamma)$ is non-decreasing. It follows that for $\gamma > \frac{\frac{\mu}{\tau} - 1}{\max_{i \in [n]} \lambda_{\max}(\mathbf{A}_i)}$, $\bar{g}_1(\gamma)$ is non-decreasing in $\gamma$ (up to a constant factor), and we arrive at the same conclusions as in the case of $\bar{g}_2(\gamma)$: $\bar{g}_1(\gamma)$ is minimized by $\gamma \in \left[\frac{1}{\max_{i \in [n]} \lambda_{\max}(\mathbf{A}_i)}, \frac{\frac{\mu}{\tau} - 1}{\max_{i \in [n]} \lambda_{\max}(\mathbf{A}_i)}\right]$ if $\frac{\mu}{\tau} \geq 2$ and by $\gamma \in \left[0, \max\left\{0, \frac{\frac{\mu}{\tau} - 1}{\max_{i \in [n]} \lambda_{\max}(\mathbf{A}_i)}\right\}\right]$ if $\frac{\mu}{\tau} < 2$.

On the other hand, by following a similar argument as in the proof of Theorem 4.3, the expression

$$\frac{1}{\lambda_{\min}^+(\gamma)} \frac{\max_{i \in [n]} \lambda_{\max}(\mathbf{A}_i)}{1 + \gamma \max_{i \in [n]} \lambda_{\max}(\mathbf{A}_i)} = \frac{\frac{\gamma \max_{i \in [n]} \lambda_{\max}(\mathbf{A}_i)}{1 + \gamma \max_{i \in [n]} \lambda_{\max}(\mathbf{A}_i)}}{\min_{\|x\|=1, x \in \ker(\mathbf{A})^\perp} \frac{1}{n} \sum_{i=1}^n x^\top \mathbf{Q}_i \left[\frac{\gamma \lambda_j(\mathbf{A}_i)}{1 + \gamma \lambda_j(\mathbf{A}_i)}\right]_{jj} \mathbf{Q}_i^\top x}$$

is approximately constant for $\gamma \geq \frac{1}{\min_{i \in [n]} \lambda_{\min}^+(\mathbf{A}_i)}$. This implies that $\bar{g}_1(\gamma)$ is increasing, and hence the intervals can be bounded above by $\frac{1}{\min_{i \in [n]} \lambda_{\min}^+(\mathbf{A}_i)}$.

The conclusion follows from the fact that the minimum of a convex combination of $\bar{g}_1(\gamma)$ and $\bar{g}_2(\gamma)$ must lie within the same interval. $\qquad \square$

*Remark* C.5. The only term in (35) and (38) that depends on $S$ is

$$L_{\gamma,S} := \frac{n - S}{S(n-1)} \frac{\max_{i \in [n]} \lambda_{\max}(\mathbf{A}_i)}{1 + \gamma \max_{i \in [n]} \lambda_{\max}(\mathbf{A}_i)} + \frac{n(S-1)}{S(n-1)} \lambda_{\max}(\gamma).$$

Recall that $\lambda_{\max}(\gamma)$ is the largest eigenvalue of the matrix $\mathbf{M} = \frac{1}{n} \sum_{i=1}^n \frac{1}{2\gamma}(\mathbf{I} - (\mathbf{I} + \gamma \mathbf{A}_i)^{-1})$, so

$$\lambda_{\max}(\gamma) \leq \frac{1}{\gamma} \frac{1}{n} \sum_{i=1}^n \frac{\lambda_{\max}(\mathbf{A}_i)\gamma}{\lambda_{\max}(\mathbf{A}_i)\gamma + 1} \leq \frac{\max_{i \in [n]} \lambda_{\max}(\mathbf{A}_i)}{\max_{i \in [n]} \lambda_{\max}(\mathbf{A}_i)\gamma + 1}.$$

As a result, since $\frac{n-S}{S(n-1)}$ is decreasing, and $\frac{n(S-1)}{S(n-1)}$ is increasing in $S$, both the iteration and time complexities are increasing functions of $S$. This underscores the advantage of involving a larger number of clients in the training process.

## C.4 LEMMAS

**Lemma C.6.** *Let $x_k$ be the iterates of SGD applied to the problem $\min_x \left\{f(x) := \frac{1}{n} \sum_{i=1}^n \left(\frac{1}{2} x^T \mathbf{B}_i x + c_i^T x + d_i\right)\right\}$, where the matrices $\mathbf{B}_i$ are symmetric. Then $x_k - \Pi(x_k) \in \text{range}(\frac{1}{n} \sum_{i=1}^n \mathbf{B}_i)$ for all $k$.*

*Proof.* By definition of $\Pi$, we have

$$\Pi(x) = x - \left(\frac{1}{n} \sum_{i=1}^n \mathbf{B}_i\right)^\dagger \left(\frac{1}{n} \sum_{i=1}^n (\mathbf{B}_i x + c_i)\right).$$

Hence, using the identity $\mathbf{A}^\top (\mathbf{A}\mathbf{A}^\top)^\dagger = \mathbf{A}^\dagger$,

$$x_k - \Pi(x_k) = \left(\frac{1}{n}\sum_{i=1}^n \mathbf{B}_i\right)^\dagger \left(\frac{1}{n}\sum_{i=1}^n (\mathbf{B}_i x_k + c_i)\right)$$

$$= \left(\frac{1}{n}\sum_{i=1}^n \mathbf{B}_i\right)^T \left(\left(\frac{1}{n}\sum_{i=1}^n \mathbf{B}_i\right)\left(\frac{1}{n}\sum_{i=1}^n \mathbf{B}_i\right)^T\right)^\dagger \left(\frac{1}{n}\sum_{i=1}^n \mathbf{B}_i \mathbb{E}\left[x_k\right] - c_i\right),$$

and so, by symmetry of $\mathbf{B}_i$, we get $x_k - \Pi(x_k) \in \mathrm{range}(\frac{1}{n}\sum_{i=1}^n \mathbf{B}_i)$. $\qquad\square$

**Lemma C.7.** *Let* $\mathbf{M} := \frac{1}{n}\sum_{i=1}^n \frac{1}{\gamma}(\mathbf{I} - (\gamma\mathbf{A}_i + \mathbf{I})^{-1})$, *where* $\gamma > 0$ *and* $\mathbf{A}_i \in \mathrm{Sym}_+^d$ *for all* $i \in [n]$. *The smallest positive eigenvalue of* $\mathbf{M}$ *is given by*

$$\lambda_{\min}^+(\gamma) := \min_{\|x\|=1, x\in(\ker \mathbf{A})^\perp} \frac{1}{n}\sum_{i=1}^n x^\top \mathbf{Q}_i \left[\frac{\lambda_j(\mathbf{A}_i)}{1 + \gamma\lambda_j(\mathbf{A}_i)}\right]_{jj} \mathbf{Q}_i^\top x,$$

*where* $\mathbf{A} = \frac{1}{n}\sum_{i=1}^n \mathbf{A}_i$ *and* $[b_j]_{jj}$ *denotes a diagonal matrix with* $b_j$ *as the jth entry.*

*Proof.* First, observe that the matrices $\mathbf{I} - (\gamma\mathbf{A}_i + \mathbf{I})^{-1}$ are symmetric, and their eigenvalues are given by

$$1 - \frac{1}{1 + \gamma\lambda_j(\mathbf{A}_i)} = \frac{\gamma\lambda_j(\mathbf{A}_i)}{1 + \gamma\lambda_j(\mathbf{A}_i)}.$$

Consequently, $\mathbf{M}$ is a sum of symmetric positive definite matrices and is therefore also symmetric positive definite. We now claim that

$$\lambda_{\min}^+(\gamma) = \min_{\|x\|=1, x\in(\ker \mathbf{M})^\perp} x^\top \mathbf{M} x.$$

First, choosing $x$ to be a multiple of the eigenvector of $\mathbf{M}$ corresponding to $\lambda_{\min}^+(\gamma)$, we see that

$$\lambda_{\min}^+(\gamma) \geq \min_{\|x\|=1, x\in(\ker \mathbf{M})^\perp} x^\top \mathbf{M} x.$$

To establish the reverse inequality, let $\{e_i\}$ be an orthonormal eigenbasis of $\mathbf{M}$ and let $x$ be such that $\|x\| = 1$. Then we can write $x = \sum_{i=1}^d \alpha_i e_i$, and since $x \in (\ker \mathbf{M})^\perp$, all coefficients corresponding to an eigenvalue 0 vanish. Thus

$$x^\top \mathbf{M} x = \sum_{i=1}^d \alpha_i^2 \lambda_i \geq \lambda_{\min}^+(\gamma)\sum_{i=1}^d \alpha_i^2 = \lambda_{\min}^+(\gamma)\|x\|^2 = \lambda_{\min}^+(\gamma).$$

This proves that

$$\lambda_{\min}^+(\gamma) = \min_{\|x\|=1, x\in(\ker \mathbf{M})^\perp} x^\top \mathbf{M} x.$$

Now, each matrix $\mathbf{I} - (\gamma\mathbf{A}_i + \mathbf{I})^{-1}$ can be decomposed as $\mathbf{Q}_i \left[\frac{\lambda_j(\mathbf{A}_i)}{1+\gamma\lambda_j(\mathbf{A}_i)}\right]_{jj} \mathbf{Q}_i^\top$. Therefore what remains to be proven is that $\ker(\mathbf{M}) = \ker(\mathbf{A})$. To this end, take $x \in \ker(\mathbf{M})$. Then $\mathbf{M}x = 0$, and since $\gamma > 0$, we have

$$\frac{1}{n}\sum_{i=1}^n (\gamma\mathbf{A}_i + \mathbf{I})^{-1}x = x. \tag{39}$$

Next, observe that

$$\lambda_{\max}\left(\frac{1}{n}\sum_{i=1}^n (\gamma\mathbf{A}_i + \mathbf{I})^{-1}\right) \leq \frac{1}{n}\sum_{i=1}^n \lambda_{\max}\left((\gamma\mathbf{A}_i + \mathbf{I})^{-1}\right) = \frac{1}{n}\sum_{i=1}^n \frac{1}{1 + \gamma\lambda_{\min}(\mathbf{A}_i)}.$$

Now consider two cases.

First, if there exists $j \in [n]$ such that $\lambda_{\min}(\mathbf{A}_j) > 0$, the above upper bound is strictly less than 1. This implies that there exists no nonzero $x$ that satisfies equation (39), and so $\ker(\mathbf{M}) = \{0\}$. But $\lambda_{\min}(\mathbf{A}_j) > 0$ also implies that

$$\lambda_{\min}\left(\frac{1}{n}\sum_{i=1}^{n}\mathbf{A}_i\right) \geq \frac{1}{n}\sum_{i=1}^{n}\lambda_{\min}(\mathbf{A}_i) > 0.$$

As a result, $\ker(\mathbf{M}) = \{0\} = \ker(\mathbf{A})$.

Now, let us suppose that $\lambda_{\min}(\mathbf{A}_i) = 0$ for all $i \in [n]$. In this case, $\lambda_{\max}\left((\gamma\mathbf{A}_i + \mathbf{I})^{-1}\right) = 1$ for all $i \in [n]$. Since all matrices $(\gamma\mathbf{A}_i + \mathbf{I})^{-1}$ are symmetric positive definite with maximum eigenvalue equal to 1, it follows that equation (39) holds if and only if

$$(\gamma\mathbf{A}_i + \mathbf{I})^{-1}x = x$$

for all $i \in [n]$. This is equivalent to $\gamma\mathbf{A}_i x = 0$ for all $i \in [n]$. Since $\gamma > 0$ and $\mathbf{A}_i$ are symmetric positive definite, we can, in turn, equivalently express it as $\mathbf{A}x = \frac{1}{n}\sum_{i=1}^{n}\mathbf{A}_i x = 0$.

Consequently, (39) holds (i.e., $x \in \ker(\mathbf{M})$) if and only if $x \in \ker(\mathbf{A})$ and hence $\ker(\mathbf{A}) = \ker(\mathbf{M})$.

The final expression in the statement of the Lemma follows from eigendecomposition (see Fact 2).
$\square$

## D FedExProx UNDER PŁ CONDITION

**Theorem D.1.** *Let Assumptions 1.1, 1.2, 1.3, and 7.1 hold, and assume that $\alpha\gamma \leq \frac{2}{L_{\gamma,S}}$. Then the iterates of Algorithm 2 satisfy*

$$\mathbb{E}\left[\|x_K - \Pi(x_K)\|^2\right] \leq \left(1 - \alpha\gamma\left(2 - \alpha\gamma L_{\gamma,S}\right)\frac{\mu_\gamma^+}{2}\right)^K \mathbb{E}\left[\|x_0 - \Pi(x_0)\|^2\right]$$

*and hence*

$$\mathbb{E}\left[f(x_K) - f(\Pi(x_K))\right] \leq \frac{L}{2}\left(1 - \alpha\gamma\left(2 - \alpha\gamma L_{\gamma,S}\right)\frac{\mu_\gamma^+}{2}\right)^K \mathbb{E}\left[\|x_0 - \Pi(x_0)\|^2\right],$$

*where $L_{\gamma,S} := \left(\frac{n-S}{S(n-1)}\frac{L_{\max}}{1+\gamma L_{\max}} + \frac{n(S-1)}{S(n-1)}L_\gamma\right)$. For the optimal choice of stepsize and extrapolation parameter $\alpha\gamma = 1/L_{\gamma,S}$, these rates become*

$$\mathbb{E}\left[\|x_K - \Pi(x_K)\|^2\right] \leq \left(1 - \frac{\mu_\gamma^+}{2L_{\gamma,S}}\right)^K \mathbb{E}\left[\|x_0 - \Pi(x_0)\|^2\right]$$

*and*

$$\mathbb{E}\left[f(x_K) - f(\Pi(x_K))\right] \leq \frac{L}{2}\left(1 - \frac{\mu_\gamma^+}{2L_{\gamma,S}}\right)^K \mathbb{E}\left[\|x_0 - \Pi(x_0)\|^2\right].$$

*Remark* D.2. The above theorem proves Theorem 7.5. Furthermore, by setting $S = n$, we recover the result from Theorem 7.2.

*Proof.* The proof closely follows the proof of Theorem 3 from Li et al. (2024). Recall from Section B that Algorithm 2 is equivalent to SGD for minimizing $M^\gamma(x) := \frac{1}{n}\sum_{i=1}^n \nabla M_{f_i}^\gamma(x)$ with stepsize $\alpha\gamma$, and its updates can be written as

$$x_{k+1} = x_k - \alpha\gamma\frac{1}{S}\sum_{i\in\mathcal{S}_k}\nabla M_{f_i}^\gamma(x_k).$$

Then, for any $x_* \in \mathcal{X}_*$

$$\mathbb{E}_k\left[\|x_{k+1} - x_*\|^2\right]$$

$$= \|x_k - x_*\|^2 - 2\alpha\gamma\left\langle x_k - x_*, \mathbb{E}_k\left[\frac{1}{S}\sum_{i\in\mathcal{S}_k}\nabla M_{f_i}^\gamma(x_k)\right]\right\rangle + \alpha^2\gamma^2\mathbb{E}_k\left[\left\|\frac{1}{S}\sum_{i\in\mathcal{S}_k}\nabla M_{f_i}^\gamma(x_k)\right\|^2\right]$$

$$\overset{(A.8)}{=} \|x_k - x_*\|^2 - 2\alpha\gamma\left\langle x_k - x_*, \nabla M^\gamma(x_k) - \nabla M^\gamma(x_*)\right\rangle + \alpha^2\gamma^2\mathbb{E}_k\left[\left\|\frac{1}{S}\sum_{i\in\mathcal{S}_k}\nabla M_{f_i}^\gamma(x_k)\right\|^2\right]$$

$$= \|x_k - x_*\|^2 - 2\alpha\gamma\left(D_{M^\gamma}(x_k, x_*) + D_{M^\gamma}(x_*, x_k)\right) + \alpha^2\gamma^2\mathbb{E}_k\left[\left\|\frac{1}{S}\sum_{i\in\mathcal{S}_k}\nabla M_{f_i}^\gamma(x_k)\right\|^2\right], \quad (40)$$

where $D_{M^\gamma}(x,y) := M^\gamma(x) - M^\gamma(y) - \langle\nabla M^\gamma(y), x - y\rangle$. Next, applying Lemma A.12 with $\mathbf{B}_i = \nabla M_{f_i}^\gamma(x_k) - \nabla M_{f_i}^\gamma(x_*) \in \mathbb{R}^{d\times 1}$, the last term in the above inequality can be written as

$$\mathbb{E}_k\left[\left\|\frac{1}{S}\sum_{i\in\mathcal{S}_k}\nabla M_{f_i}^\gamma(x_k)\right\|^2\right] \overset{(A.5)}{=} \mathbb{E}_k\left[\left\|\frac{1}{S}\sum_{i\in\mathcal{S}_k}\left(\nabla M_{f_i}^\gamma(x_k) - \nabla M_{f_i}^\gamma(x_*)\right)\right\|^2\right]$$

$$\overset{(A.12)}{=} \frac{n-S}{S(n-1)}\frac{1}{n}\sum_{i=1}^n\left\|\nabla M_{f_i}^\gamma(x_k) - \nabla M_{f_i}^\gamma(x_*)\right\|^2$$

$$+ \frac{n(S-1)}{S(n-1)}\left\|\frac{1}{n}\sum_{i=1}^n\left(\nabla M_{f_i}^\gamma(x_k) - \nabla M_{f_i}^\gamma(x_*)\right)\right\|^2.$$

Looking at the first term of the inequality above, using convexity and smoothness of the functions $M_{f_i}^\gamma$ (Lemmas A.4 and A.6), we have

$$\frac{1}{n} \sum_{i=1}^n \left\| \nabla M_{f_i}^\gamma(x_k) - \nabla M_{f_i}^\gamma(x_*) \right\|^2 \overset{\text{(A.14)}}{\leq} \frac{1}{n} \sum_{i=1}^n \frac{L_i}{1 + \gamma L_i} \left( D_{M_{f_i}^\gamma}(x_k, x_*) + D_{M_{f_i}^\gamma}(x_*, x_k) \right)$$

$$\leq \frac{L_{\max}}{1 + \gamma L_{\max}} \frac{1}{n} \sum_{i=1}^n \left( D_{M_{f_i}^\gamma}(x_k, x_*) + D_{M_{f_i}^\gamma}(x_*, x_k) \right)$$

$$= \frac{L_{\max}}{1 + \gamma L_{\max}} \left( D_{M^\gamma}(x_k, x_*) + D_{M^\gamma}(x_*, x_k) \right).$$

Next, since by Lemma A.7, the function $M^\gamma$ is convex and smooth, the second term can be bounded as

$$\left\| \frac{1}{n} \sum_{i=1}^n \left( \nabla M_{f_i}^\gamma(x_k) - \nabla M_{f_i}^\gamma(x_*) \right) \right\|^2 = \left\| \nabla M^\gamma(x_k) - \nabla M^\gamma(x_*) \right\|^2$$

$$\overset{\text{(A.14)}}{\leq} L_\gamma \left( D_{M^\gamma}(x_k, x_*) + D_{M^\gamma}(x_*, x_k) \right).$$

Applying these bounds in (40) gives

$$\mathbb{E}_k \left[ \|x_{k+1} - x_*\|^2 \right]$$

$$\leq \|x_k - x_*\|^2 - 2\alpha\gamma \left( D_{M^\gamma}(x_k, x_*) + D_{M^\gamma}(x_*, x_k) \right)$$

$$+ \alpha^2 \gamma^2 \underbrace{\left( \frac{n - S}{S(n-1)} \frac{L_{\max}}{1 + \gamma L_{\max}} + \frac{n(S-1)}{S(n-1)} L_\gamma \right)}_{:= L_{\gamma,S}} \left( D_{M^\gamma}(x_k, x_*) + D_{M^\gamma}(x_*, x_k) \right)$$

$$= \|x_k - x_*\|^2 - \alpha\gamma \left( 2 - \alpha\gamma L_{\gamma,S} \right) \left( D_{M^\gamma}(x_k, x_*) + D_{M^\gamma}(x_*, x_k) \right).$$

By Lemma A.7, $M^\gamma$ is convex and smooth, so $D_{M^\gamma}(x_*, x_k) \geq 0$, and by Assumption 7.1 and Lemma A.13, we have

$$D_{M^\gamma}(x_k, x_*) = M^\gamma(x_k) - M^\gamma(x_*) \geq \frac{\mu_\gamma^+}{2} \|x_k - \Pi(x_k)\|^2 .$$

Therefore, for $\alpha\gamma L_{\gamma,S} \leq 2$

$$\mathbb{E}_k \left[ \|x_{k+1} - x_*\|^2 \right] \leq \|x_k - x_*\|^2 - \alpha\gamma \left( 2 - \alpha\gamma L_{\gamma,S} \right) \left( D_{M^\gamma}(x_k, x_*) + D_{M^\gamma}(x_*, x_k) \right)$$

$$\leq \|x_k - x_*\|^2 - \alpha\gamma \left( 2 - \alpha\gamma L_{\gamma,S} \right) \frac{\mu_\gamma^+}{2} \|x_k - \Pi(x_k)\|^2 .$$

Taking expectation and letting $x_* = \Pi(x_k)$ gives

$$\mathbb{E} \left[ \|x_{k+1} - \Pi(x_{k+1})\|^2 \right] \leq \mathbb{E} \left[ \|x_{k+1} - \Pi(x_k)\|^2 \right]$$

$$\leq \mathbb{E} \left[ \|x_k - \Pi(x_k)\|^2 \right] - \alpha\gamma \left( 2 - \alpha\gamma L_{\gamma,S} \right) \frac{\mu_\gamma^+}{2} \mathbb{E} \left[ \|x_k - \Pi(x_k)\|^2 \right] .$$

Unrolling the recurrence, we obtain the first result

$$\mathbb{E} \left[ \|x_K - \Pi(x_K)\|^2 \right] \leq \left( 1 - \alpha\gamma \left( 2 - \alpha\gamma L_{\gamma,S} \right) \frac{\mu_\gamma^+}{2} \right)^K \mathbb{E} \left[ \|x_0 - \Pi(x_0)\|^2 \right] .$$

Lastly, using $L$–smoothness of $f$, it follows that

$$\mathbb{E} \left[ f(x_K) - f(\Pi(x_K)) \right] \leq \frac{L}{2} \mathbb{E} \left[ \|x_K - \Pi(x_K)\|^2 \right]$$

$$\leq \frac{L}{2} \left( 1 - \alpha\gamma \left( 2 - \alpha\gamma L_{\gamma,S} \right) \frac{\mu_\gamma^+}{2} \right)^K \mathbb{E} \left[ \|x_0 - \Pi(x_0)\|^2 \right] .$$

Substituting $\alpha\gamma = {}^1/_{L_{\gamma,S}}$, which minimizes the expression $\alpha\gamma \left( 2 - \alpha\gamma L_{\gamma,S} \right)$, finishes the proof. $\quad\square$

## E    FedExProx WITH INEXACT COMPUTATIONS

In practice, solving (3) exactly is often infeasible, and we can only find a vector $\text{prox}_{\gamma f}^{\delta}(x)$ such that

$$\left\|\text{prox}_{\gamma f}^{\delta}(x) - \text{prox}_{\gamma f}(x)\right\|^2 \leq \delta,$$

where $\delta$ represents the accuracy of the approximate solution to (3). As a result, we can only calculate an inexact gradient of the Moreau envelope, defined as

$$\nabla M_f^{\gamma,\delta}(x) := \frac{1}{\gamma}(x - \text{prox}_{\gamma f}^{\delta}(x)).$$

One can easily show that

$$\left\|\nabla M_f^{\gamma,\delta}(x) - \nabla M_f^{\gamma}(x)\right\|^2 \leq \frac{\delta}{\gamma^2}.$$

With these inexact updates, Algorithm 1 iterates

$$x_{k+1} = x_k + \alpha_k \left(\frac{1}{n}\sum_{i=1}^{n} \text{prox}_{\gamma f_i}^{\delta}(x_k) - x_k\right) = x_k - \alpha_k \gamma \underbrace{\frac{1}{n}\sum_{i=1}^{n} \nabla M_{f_i}^{\gamma,\delta}(x_k)}_{:=\nabla M^{\gamma,\delta}(x_k)}, \qquad (41)$$

where

$$\left\|\nabla M^{\gamma,\delta}(x) - \nabla M^{\gamma}(x)\right\|^2 = \left\|\frac{1}{n}\sum_{i=1}^{n}\left(\nabla M_{f_i}^{\gamma,\delta}(x) - \nabla M_{f_i}^{\gamma}(x)\right)\right\|^2$$

$$= \frac{1}{\gamma^2}\left\|\frac{1}{n}\sum_{i=1}^{n}\left(\text{prox}_{\gamma f}^{\delta}(x) - \text{prox}_{\gamma f}(x)\right)\right\|^2$$

$$\leq \frac{1}{\gamma^2}\frac{1}{n}\sum_{i=1}^{n}\left\|\text{prox}_{\gamma f}^{\delta}(x) - \text{prox}_{\gamma f}(x)\right\|^2 \leq \frac{\delta}{\gamma^2}. \qquad (42)$$

With this in mind, we proceed to our analysis.

**Theorem E.1.** *Consider the inexact* FedExProx *method in the full participation setting defined in* (41). *Let Assumptions 1.1, 1.2, 1.3, and 7.1 hold, and set* $\alpha\gamma = \frac{1}{2L_\gamma}$. *Then, the iterates of the algorithm satisfy*

$$\|x_K - \Pi(x_K)\|^2 \leq \left(1 - \frac{\mu_\gamma^+}{8L_\gamma}\right)^K \|x_0 - \Pi(x_0)\|^2 + \frac{20\delta}{\left(\mu_\gamma^+\right)^2\gamma^2},$$

*and hence*

$$f(x_K) - f(\Pi(x_K)) \leq \frac{L}{2}\left(1 - \frac{\mu_\gamma^+}{8L_\gamma}\right)^K \|x_0 - \Pi(x_0)\|^2 + \frac{10L\delta}{\left(\mu_\gamma^+\right)^2\gamma^2}. \qquad (43)$$

*Remark* E.2. Note that the price one has to pay for inexactness is minimal. To illustrate this, suppose that clients solve the local problems using GD. Then, taking

$$\delta = \frac{\left(\mu_\gamma^+\right)^2\gamma^2\varepsilon}{20L},$$

we ensure that the last term in (43) is small:

$$\frac{10L\delta}{\left(\mu_\gamma^+\right)^2\gamma^2} \leq \frac{\varepsilon}{2}.$$

Since problem (3) is strongly convex, the local complexity is proportional to

$$\mathcal{O}\left(\kappa \times \log\frac{1}{\delta}\right) = \mathcal{O}\left(\kappa \times \log\left(\frac{20L}{\left(\mu_\gamma^+\right)^2\gamma^2\varepsilon}\right)\right),$$

where $\kappa \gg 1$ is the condition number. In practice, this logarithmic factor can be ignored and treated as a constant.

*Proof of Theorem E.1.* The updates can be written as

$$x_{k+1} = x_k - \alpha\gamma \nabla M^{\gamma,\delta}(x_k).$$

Thus

$$\|x_{k+1} - \Pi(x_{k+1})\|^2 \leq \|x_{k+1} - \Pi(x_k)\|^2$$
$$= \|x_k - \Pi(x_k)\|^2 - 2\alpha\gamma \langle x_k - \Pi(x_k), \nabla M^{\gamma,\delta}(x_k)\rangle + \alpha^2\gamma^2 \left\|\nabla M^{\gamma,\delta}(x_k)\right\|^2$$
$$= \|x_k - \Pi(x_k)\|^2 - 2\alpha\gamma \langle x_k - \Pi(x_k), \nabla M^{\gamma}(x_k)\rangle$$
$$\quad - 2\alpha\gamma \langle x_k - \Pi(x_k), \nabla M^{\gamma,\delta}(x_k) - \nabla M^{\gamma}(x_k)\rangle$$
$$\quad + \alpha^2\gamma^2 \left\|\nabla M^{\gamma}(x_k) + (\nabla M^{\gamma,\delta}(x_k) - \nabla M^{\gamma}(x_k))\right\|^2.$$

Using Jensen's inequality, we get

$$\|x_{k+1} - \Pi(x_{k+1})\|^2 \leq \|x_k - \Pi(x_k)\|^2 - 2\alpha\gamma \langle x_k - \Pi(x_k), \nabla M^{\gamma}(x_k)\rangle$$
$$\quad - 2\alpha\gamma \langle x_k - \Pi(x_k), \nabla M^{\gamma,\delta}(x_k) - \nabla M^{\gamma}(x_k)\rangle$$
$$\quad + 2\alpha^2\gamma^2 \|\nabla M^{\gamma}(x_k)\|^2 + 2\alpha^2\gamma^2 \left\|\nabla M^{\gamma,\delta}(x_k) - \nabla M^{\gamma}(x_k)\right\|^2. \quad (44)$$

Following the same steps as in the proof of Theorem D.1 gives

$$\|x_k - \Pi(x_k)\|^2 - 2\alpha\gamma \langle x_k - \Pi(x_k), \nabla M^{\gamma}(x_k)\rangle + 2\alpha^2\gamma^2 \|\nabla M^{\gamma}(x_k)\|^2$$
$$\leq \left(1 - \alpha\gamma(1 - \alpha\gamma L_{\gamma})\mu_{\gamma}^{+}\right) \|x_k - \Pi(x_k)\|^2$$
$$\leq \left(1 - \alpha\gamma \frac{\mu_{\gamma}^{+}}{2}\right) \|x_k - \Pi(x_k)\|^2$$

for $\alpha\gamma \leq \frac{1}{2L_{\gamma}}$, and hence

$$\|x_{k+1} - \Pi(x_{k+1})\|^2 \overset{(44)}{\leq} \left(1 - \alpha\gamma \frac{\mu_{\gamma}^{+}}{2}\right) \|x_k - \Pi(x_k)\|^2$$
$$\quad - 2\alpha\gamma \langle x_k - \Pi(x_k), \nabla M^{\gamma,\delta}(x_k) - \nabla M^{\gamma}(x_k)\rangle$$
$$\quad + 2\alpha^2\gamma^2 \left\|\nabla M^{\gamma,\delta}(x_k) - \nabla M^{\gamma}(x_k)\right\|^2.$$

Young's inequality yields

$$\|x_{k+1} - \Pi(x_{k+1})\|^2 \leq \left(1 - \alpha\gamma \frac{\mu_{\gamma}^{+}}{2}\right) \|x_k - \Pi(x_k)\|^2 + \alpha\gamma\lambda \|x_k - \Pi(x_k)\|^2$$
$$\quad + \frac{\alpha\gamma}{\lambda} \left\|\nabla M^{\gamma,\delta}(x_k) - \nabla M^{\gamma}(x_k)\right\|^2 + 2\alpha^2\gamma^2 \left\|\nabla M^{\gamma,\delta}(x_k) - \nabla M^{\gamma}(x_k)\right\|^2,$$

where $\lambda > 0$ is a free parameter. Taking $\lambda = \frac{\mu_{\gamma}^{+}}{4}$, we obtain

$$\|x_{k+1} - \Pi(x_{k+1})\|^2 \leq \left(1 - \alpha\gamma \frac{\mu_{\gamma}^{+}}{4}\right) \|x_k - \Pi(x_k)\|^2 + \frac{4\alpha\gamma}{\mu_{\gamma}^{+}} \left\|\nabla M^{\gamma,\delta}(x_k) - \nabla M^{\gamma}(x_k)\right\|^2$$
$$\quad + 2\alpha^2\gamma^2 \left\|\nabla M^{\gamma,\delta}(x_k) - \nabla M^{\gamma}(x_k)\right\|^2$$
$$\leq \left(1 - \alpha\gamma \frac{\mu_{\gamma}^{+}}{4}\right) \|x_k - \Pi(x_k)\|^2 + \frac{5\alpha\gamma}{\mu_{\gamma}^{+}} \left\|\nabla M^{\gamma,\delta}(x_k) - \nabla M^{\gamma}(x_k)\right\|^2,$$

where the last inequality due to $\alpha\gamma \leq \frac{1}{2L_{\gamma}}$ and $L_{\gamma} \geq \mu_{\gamma}^{+}$. Lastly, due to (42), we have

$$\|x_{k+1} - \Pi(x_{k+1})\|^2 \leq \left(1 - \alpha\gamma \frac{\mu_{\gamma}^{+}}{4}\right) \|x_k - \Pi(x_k)\|^2 + \frac{5\alpha\delta}{\mu_{\gamma}^{+}\gamma}.$$

Unrolling the recursion, one can show that

$$\|x_K - \Pi(x_K)\|^2 \leq \left(1 - \alpha\gamma \frac{\mu_{\gamma}^{+}}{4}\right)^K \|x_0 - \Pi(x_0)\|^2 + \frac{20\delta}{\left(\mu_{\gamma}^{+}\right)^2 \gamma^2},$$

and hence

$$
\begin{aligned}
f(x_K) - f(\Pi(x_K)) &\leq \frac{L}{2} \left\| x_K - \Pi(x_K) \right\|^2 \\
&\leq \frac{L}{2} \left( 1 - \alpha\gamma \frac{\mu_\gamma^+}{4} \right)^K \left\| x_0 - \Pi(x_0) \right\|^2 + \frac{10L\delta}{\left( \mu_\gamma^+ \right)^2 \gamma^2}.
\end{aligned}
$$

$\square$

# F    ADAPTIVE EXTRAPOLATION

The optimal extrapolation parameter values in Theorems 4.1, 5.1, 7.2 and 7.5 depend on $L_\gamma$ (or $L_{\gamma,S}$). This dependency can be avoided by employing *adaptive* extrapolation strategies (Horváth et al., 2022). Specifically, we analyze methods based on gradient diversity (FedExProx-GraDS) and the Polyak stepsize (FedExProx-StoPS).

## F.1    FULL PARTICIPATION CASE

**Theorem 6.1.** *Fix any $\gamma > 0$ and consider solving non-strongly convex quadratic optimization problem* (1), *where $f_i(x) = \frac{1}{2} x^\top \mathbf{A}_i x - b_i^\top x$ for all $i \in [n]$, with $\mathbf{A}_i \in \mathrm{Sym}_+^d$ and $b_i \in \mathbb{R}^d$. Let Assumption 1.3 hold and consider two adaptive extrapolation strategies:*

    *1. (FedExProx-GraDS) Set*

$$\alpha_k = \alpha_k^{\mathsf{GraDS}}(x_k) := \frac{\frac{1}{n}\sum_{i=1}^n \left\|\nabla M_{f_i}^\gamma(x_k)\right\|^2}{\left\|\frac{1}{n}\sum_{i=1}^n \nabla M_{f_i}^\gamma(x_k)\right\|^2} \geq 1.$$

    *Then, the iterates of Algorithm 1 satisfy*

$$\|x_K - \Pi(x_K)\|^2 \leq \left(1 - \min_{k=0,\ldots,K-1} \alpha_k \gamma \frac{2+\gamma L_{\max}}{1+\gamma L_{\max}} \mu_\gamma^+\right)^K \|x_0 - \Pi(x_0)\|^2. \quad (15)$$

    *2. (FedExProx-StoPS) Set*

$$\alpha_k = \alpha_k^{\mathsf{StoPS}}(x_k) := \frac{\frac{1}{n}\sum_{i=1}^n \left(M_{f_i}^\gamma(x_k) - \inf M_{f_i}^\gamma\right)}{\gamma \left\|\frac{1}{n}\sum_{i=1}^n \nabla M_{f_i}^\gamma(x_k)\right\|^2} \geq \frac{1}{2\gamma L_\gamma}.$$

    *Then, the iterates of Algorithm 1 satisfy*

$$\|x_K - \Pi(x_K)\|^2 \leq \left(1 - \frac{3}{2} \min_{k=0,\ldots,K-1} \alpha_k \gamma \mu_\gamma^+\right)^K \|x_0 - \Pi(x_0)\|^2. \quad (16)$$

*Proof.* We start with the standard decomposition

$$
\begin{aligned}
\|x_{k+1} - \Pi(x_{k+1})\|^2 &\leq \|x_{k+1} - \Pi(x_k)\|^2 \\
&= \|x_k - \Pi(x_k)\|^2 - 2\alpha_k\gamma \left\langle \frac{1}{n}\sum_{i=1}^n \nabla M_{f_i}^\gamma(x_k), x_k - \Pi(x_k) \right\rangle \\
&\quad + \alpha_k^2\gamma^2 \left\|\frac{1}{n}\sum_{i=1}^n \nabla M_{f_i}^\gamma(x_k)\right\|^2 \\
&\overset{(23)}{=} \|x_k - \Pi(x_k)\|^2 - 2\alpha_k\gamma(x_k - \Pi(x_k))^\top \mathbf{M}(x_k - \Pi(x_k)) \\
&\quad + \alpha_k^2\gamma^2 \left\|\frac{1}{n}\sum_{i=1}^n \nabla M_{f_i}^\gamma(x_k)\right\|^2. \quad (45)
\end{aligned}
$$

Now, let us consider the two adaptive extrapolation strategies.

    1. (FedExProx-GraDS) In this case, the last term of (45) can be rewritten as

$$
\begin{aligned}
\alpha_k^2\gamma^2 &\left\|\frac{1}{n}\sum_{i=1}^n \nabla M_{f_i}^\gamma(x_k)\right\|^2 \\
&= \alpha_k\gamma^2 \frac{1}{n}\sum_{i=1}^n \left\|\nabla M_{f_i}^\gamma(x_k)\right\|^2 \\
&\overset{(23)}{=} \alpha_k\gamma^2 \frac{1}{n}\sum_{i=1}^n \left\|\frac{1}{\gamma}\left(\mathbf{I} - (\gamma\mathbf{A}_i + \mathbf{I})^{-1}\right)(x_k - \Pi(x_k))\right\|^2
\end{aligned}
$$

$$= \alpha_k (x_k - \Pi(x_k))^\top \left( \frac{1}{n} \sum_{i=1}^n \left( \mathbf{I} - (\gamma \mathbf{A}_i + \mathbf{I})^{-1} \right)^\top \left( \mathbf{I} - (\gamma \mathbf{A}_i + \mathbf{I})^{-1} \right) \right) (x_k - \Pi(x_k))$$

$$\leq \alpha_k \max_{i \in [n]} \lambda_{\max} \left( \mathbf{I} - (\gamma \mathbf{A}_i + \mathbf{I})^{-1} \right) (x_k - \Pi(x_k))^\top \left( \frac{1}{n} \sum_{i=1}^n \left( \mathbf{I} - (\gamma \mathbf{A}_i + \mathbf{I})^{-1} \right) \right) (x_k - \Pi(x_k))$$

$$= \alpha_k \gamma \frac{\gamma \max_{i \in [n]} \lambda_{\max}(\mathbf{A}_i)}{1 + \gamma \max_{i \in [n]} \lambda_{\max}(\mathbf{A}_i)} (x_k - \Pi(x_k))^\top \mathbf{M}(x_k - \Pi(x_k)).$$

Substituting this in (45) gives

$$\|x_{k+1} - \Pi(x_{k+1})\|^2$$
$$\leq \|x_k - \Pi(x_k)\|^2 - 2\alpha_k \gamma (x_k - \Pi(x_k))^\top \mathbf{M}(x_k - \Pi(x_k))$$
$$+ \alpha_k \gamma \frac{\gamma \max_{i \in [n]} \lambda_{\max}(\mathbf{A}_i)}{1 + \gamma \max_{i \in [n]} \lambda_{\max}(\mathbf{A}_i)} (x_k - \Pi(x_k))^\top \mathbf{M}(x_k - \Pi(x_k))$$
$$= \|x_k - \Pi(x_k)\|^2 - \alpha_k \gamma \frac{2 + \gamma \max_{i \in [n]} \lambda_{\max}(\mathbf{A}_i)}{1 + \gamma \max_{i \in [n]} \lambda_{\max}(\mathbf{A}_i)} (x_k - \Pi(x_k))^\top \mathbf{M}(x_k - \Pi(x_k))$$
$$\leq \left( 1 - \alpha_k \gamma \frac{2 + \gamma \max_{i \in [n]} \lambda_{\max}(\mathbf{A}_i)}{1 + \gamma \max_{i \in [n]} \lambda_{\max}(\mathbf{A}_i)} \lambda_{\min}^+(\gamma) \right) \|x_k - \Pi(x_k)\|^2 ,$$

where the last inequality follows from the fact that $x_k - \Pi(x_k) \in \text{range}(\mathbf{M})$ (see Lemma C.6). Applying this bound iteratively for $k = 0, \ldots, K-1$ gives

$$\|x_K - \Pi(x_K)\|^2$$
$$\leq \left( 1 - \gamma \min_{k=0,\ldots,K-1} \alpha_k \frac{2 + \gamma \max_{i \in [n]} \lambda_{\max}(\mathbf{A}_i)}{1 + \gamma \max_{i \in [n]} \lambda_{\max}(\mathbf{A}_i)} \lambda_{\min}^+(\gamma) \right)^K \|x_0 - \Pi(x_0)\|^2 .$$

2. (FedExProx-StoPS) First, note that $L_\gamma$–smoothness of $M^\gamma$ implies that

$$M^\gamma(x_k) - \inf M^\gamma \geq \frac{1}{2L_\gamma} \|\nabla M^\gamma(x_k)\|^2 ,$$

and hence $\alpha_{k,S}(x_k) \geq \frac{1}{2\gamma L_\gamma}$.

Now, the last term of (45) is

$$\alpha_k^2 \gamma^2 \left\| \frac{1}{n} \sum_{i=1}^n \nabla M_{f_i}^\gamma(x_k) \right\|^2 = \alpha_k \gamma \frac{1}{n} \sum_{i=1}^n \left( M_{f_i}^\gamma(x_k) - \inf M_{f_i}^\gamma \right)$$
$$= \alpha_k \gamma \left( M^\gamma(x_k) - M^\gamma(\Pi(x_k)) \right)$$
$$\overset{(26)}{=} \frac{1}{2} \alpha_k \gamma (x_k - \Pi(x_k))^T \mathbf{M}(x_k - \Pi(x_k)).$$

Therefore, using Lemma C.6, we get

$$\|x_{k+1} - \Pi(x_{k+1})\|^2 \overset{(45)}{\leq} \|x_k - \Pi(x_k)\|^2 - 2\alpha_k \gamma (x_k - \Pi(x_k))^\top \mathbf{M}(x_k - \Pi(x_k))$$
$$+ \frac{1}{2} \alpha_k \gamma (x - \Pi(x_k))^T \mathbf{M}(x - \Pi(x_k))$$
$$\leq \left( 1 - \frac{3}{2} \alpha_k \gamma \lambda_{\min}^+(\gamma) \right) \|x_k - \Pi(x_k)\|^2 .$$

Applying this bound iteratively for $k = 0, \ldots, K-1$ gives

$$\|x_K - \Pi(x_K)\|^2 \leq \left( 1 - \frac{3}{2} \min_{k=0,\ldots,K-1} \alpha_k \gamma \lambda_{\min}^+(\gamma) \right)^K \|x_0 - \Pi(x_0)\|^2 .$$

$\square$

### F.2 PARTIAL PARTICIPATION CASE

**Theorem F.1.** *Fix any $\gamma > 0$ and consider solving non-strongly convex quadratic optimization problem* (1), *where $f_i(x) = \frac{1}{2} x^\top \mathbf{A}_i x - b_i^\top x$ for all $i \in [n]$, with $\mathbf{A}_i \in \mathrm{Sym}_+^d$ and $b_i \in \mathbb{R}^d$. Let Assumption 1.3 hold and consider two adaptive extrapolation strategies:*

    *1. (FedExProx-GraDS) Set*

$$\alpha_k = \alpha_{k,S}^{\mathsf{GraDS}}(x_k, \mathcal{S}_k) := \frac{\frac{1}{S} \sum_{i \in \mathcal{S}^k} \left\| \nabla M_{f_i}^\gamma(x_k) \right\|^2}{\left\| \frac{1}{S} \sum_{i \in \mathcal{S}^k} \nabla M_{f_i}^\gamma(x_k) \right\|^2} \geq 1.$$

    *Then, the iterates of Algorithm 2 satisfy*

$$\mathbb{E}\left[ \|x_K - \Pi(x_K)\|^2 \right] \leq \left( 1 - \inf \alpha_k \gamma \frac{2 + \gamma L_{\max}}{1 + \gamma L_{\max}} \lambda_{\min}^+(\gamma) \right)^K \|x_0 - \Pi(x_0)\|^2 , \quad (46)$$

    *where*

$$\inf \alpha_k := \inf_{x \in \mathbb{R}^d, \mathcal{S} \subseteq [n], |\mathcal{S}| = S} \alpha_{k,S}^{\mathsf{GraDS}}(x, \mathcal{S}).$$

    *2. (FedExProx-StoPS) Set*

$$\alpha_k = \alpha_{k,S}^{\mathsf{StoPS}}(x_k, \mathcal{S}_k) := \frac{\frac{1}{S} \sum_{i \in \mathcal{S}^k} \left( M_{f_i}^\gamma(x_k) - \inf M_{f_i}^\gamma \right)}{\gamma \left\| \frac{1}{n} \sum_{i \in \mathcal{S}^k} \nabla M_{f_i}^\gamma(x_k) \right\|^2}.$$

    *Then, the iterates of Algorithm 2 satisfy*

$$\mathbb{E}\left[ \|x_K - \Pi(x_K)\|^2 \right] \leq \left( 1 - \frac{3}{2} \inf \alpha_k \gamma \lambda_{\min}^+(\gamma) \right)^K \|x_0 - \Pi(x_0)\|^2 , \quad (47)$$

    *where*

$$\inf \alpha_k := \inf_{x \in \mathbb{R}^d, \mathcal{S} \subseteq [n], |\mathcal{S}| = S} \alpha_{k,S}^{\mathsf{StoPS}}(x, \mathcal{S}).$$

*Remark* F.2. In the single node setting ($S = 1$),

$$\inf_{x \in \mathbb{R}^d, \mathcal{S} \subseteq [n], |\mathcal{S}| = 1} \alpha_{k,1}^{\mathsf{GraDS}}(x, \mathcal{S}) = 1.$$

However, as more clients participate, the extrapolation parameter may exceed 1, resulting in improved performance of FedExProx-GraDS.

*Remark* F.3. In the single node setting ($S = 1$), Lemma A.4 implies that

$$\inf_{x \in \mathbb{R}^d, \mathcal{S} \subseteq [n], |\mathcal{S}| = 1} \alpha_{k,1}^{\mathsf{StoPS}}(x, \mathcal{S}) = \frac{1}{2} \frac{1 + \gamma L_{\max}}{\gamma L_{\max}}.$$

As the number of participating clients increases, the extrapolation parameter may exceed this bound, leading to better performance of FedExProx-StoPS.

*Proof of Theorem F.1.* As in the full batch case, we start with the decomposition

$$\begin{aligned}
\|x_{k+1} - \Pi(x_{k+1})\|^2 &\leq \|x_{k+1} - \Pi(x_k)\|^2 \\
&= \|x_k - \Pi(x_k)\|^2 - 2\alpha_k \gamma \left\langle \frac{1}{S} \sum_{i \in \mathcal{S}^k} \nabla M_{f_i}^\gamma(x_k), x_k - \Pi(x_k) \right\rangle \\
&\quad + \alpha_k^2 \gamma^2 \left\| \frac{1}{S} \sum_{i \in \mathcal{S}^k} \nabla M_{f_i}^\gamma(x_k) \right\|^2
\end{aligned}$$

$$\stackrel{(23)}{=} \|x_k - \Pi(x_k)\|^2 - 2\alpha_k\gamma(x_k - \Pi(x_k))^\top \mathbf{M}_k(x_k - \Pi(x_k))$$

$$+\alpha_k^2\gamma^2 \left\| \frac{1}{S}\sum_{i\in\mathcal{S}^k}\nabla M_{f_i}^\gamma(x_k) \right\|^2, \tag{48}$$

where $\mathbf{M}_k := \frac{1}{\gamma}\left(\mathbf{I} - \frac{1}{S}\sum_{i\in\mathcal{S}_k}(\gamma\mathbf{A}_i + \mathbf{I})^{-1}\right)$. Now, let us consider the two adaptive extrapolation strategies.

1. (FedExProx-GraDS) In this case, the last term of (48) can be rewritten as

$$\alpha_k^2\gamma^2 \left\| \frac{1}{S}\sum_{i\in\mathcal{S}^k}\nabla M_{f_i}^\gamma(x_k) \right\|^2$$

$$= \alpha_k\gamma^2 \frac{1}{S}\sum_{i\in\mathcal{S}^k}\left\|\nabla M_{f_i}^\gamma(x_k)\right\|^2$$

$$\stackrel{(23)}{=} \alpha_k\gamma^2\frac{1}{S}\sum_{i\in\mathcal{S}^k}\left\|\frac{1}{\gamma}\left(\mathbf{I} - (\gamma\mathbf{A}_i + \mathbf{I})^{-1}\right)(x_k - \Pi(x_k))\right\|^2$$

$$= \alpha_k(x_k - \Pi(x_k))^\top\left(\frac{1}{S}\sum_{i\in\mathcal{S}^k}\left(\mathbf{I} - (\gamma\mathbf{A}_i + \mathbf{I})^{-1}\right)^\top\left(\mathbf{I} - (\gamma\mathbf{A}_i + \mathbf{I})^{-1}\right)\right)(x_k - \Pi(x_k))$$

$$\leq \alpha_k\max_{i\in[n]}\lambda_{\max}\left(\mathbf{I} - (\gamma\mathbf{A}_i + \mathbf{I})^{-1}\right)(x_k - \Pi(x_k))^\top\left(\frac{1}{S}\sum_{i\in\mathcal{S}^k}\left(\mathbf{I} - (\gamma\mathbf{A}_i + \mathbf{I})^{-1}\right)\right)(x_k - \Pi(x_k))$$

$$= \alpha_k\gamma\frac{\gamma\max_{i\in[n]}\lambda_{\max}(\mathbf{A}_i)}{1 + \gamma\max_{i\in[n]}\lambda_{\max}(\mathbf{A}_i)}(x_k - \Pi(x_k))^\top\mathbf{M}_k(x_k - \Pi(x_k)).$$

Substituting this in (48) gives

$$\|x_{k+1} - \Pi(x_{k+1})\|^2$$

$$\leq \|x_k - \Pi(x_k)\|^2 - 2\alpha_k\gamma(x_k - \Pi(x_k))^\top\mathbf{M}_k(x_k - \Pi(x_k))$$

$$+\alpha_k\gamma\frac{\gamma\max_{i\in[n]}\lambda_{\max}(\mathbf{A}_i)}{1 + \gamma\max_{i\in[n]}\lambda_{\max}(\mathbf{A}_i)}(x_k - \Pi(x_k))^\top\mathbf{M}_k(x_k - \Pi(x_k))$$

$$= \|x_k - \Pi(x_k)\|^2 - \alpha_k\gamma\frac{2 + \gamma\max_{i\in[n]}\lambda_{\max}(\mathbf{A}_i)}{1 + \gamma\max_{i\in[n]}\lambda_{\max}(\mathbf{A}_i)}(x_k - \Pi(x_k))^\top\mathbf{M}_k(x_k - \Pi(x_k))$$

$$\leq \|x_k - \Pi(x_k)\|^2 - \inf\alpha_k\gamma\frac{2 + \gamma\max_{i\in[n]}\lambda_{\max}(\mathbf{A}_i)}{1 + \gamma\max_{i\in[n]}\lambda_{\max}(\mathbf{A}_i)}(x_k - \Pi(x_k))^\top\mathbf{M}_k(x_k - \Pi(x_k)),$$

where we view $\alpha_k$ as a function of $x$ an $\mathcal{S}$. Taking expectation conditioned on $x_k$, we obtain

$$\mathbb{E}_k\left[\|x_{k+1} - \Pi(x_{k+1})\|^2\right]$$

$$\leq \|x_k - \Pi(x_k)\|^2 - \inf\alpha_k\gamma\frac{2 + \gamma\max_{i\in[n]}\lambda_{\max}(\mathbf{A}_i)}{1 + \gamma\max_{i\in[n]}\lambda_{\max}(\mathbf{A}_i)}(x_k - \Pi(x_k))^\top\mathbb{E}_k\left[\mathbf{M}_k\right](x_k - \Pi(x_k))$$

$$= \|x_k - \Pi(x_k)\|^2 - \inf\alpha_k\gamma\frac{2 + \gamma\max_{i\in[n]}\lambda_{\max}(\mathbf{A}_i)}{1 + \gamma\max_{i\in[n]}\lambda_{\max}(\mathbf{A}_i)}(x_k - \Pi(x_k))^\top\mathbf{M}(x_k - \Pi(x_k))$$

$$\leq \left(1 - \inf\alpha_k\gamma\frac{2 + \gamma\max_{i\in[n]}\lambda_{\max}(\mathbf{A}_i)}{1 + \gamma\max_{i\in[n]}\lambda_{\max}(\mathbf{A}_i)}\lambda_{\min}^+(\gamma)\right)\|x_k - \Pi(x_k)\|^2,$$

where the last inequality follows from the fact that $x_k - \Pi(x_k) \in \text{range}(\mathbf{M})$ (see Lemma C.6). Taking full expectation and applying this bound iteratively for $k = 0, \ldots, K - 1$ gives the final result.

2. (FedExProx-StoPS) In this case, the last term of (48) is

$$\alpha_k^2 \gamma^2 \left\| \frac{1}{S} \sum_{i \in \mathcal{S}^k} \nabla M_{f_i}^{\gamma}(x_k) \right\|^2$$

$$= \alpha_k \gamma \frac{1}{S} \sum_{i \in \mathcal{S}^k} \left( M_{f_i}^{\gamma}(x_k) - \inf M_{f_i}^{\gamma} \right)$$

$$\stackrel{(24)}{=} \frac{1}{2} \alpha_k \frac{1}{S} \sum_{i \in \mathcal{S}^k} (x_k - \Pi(x_k))^T (\mathbf{I} - (\gamma \mathbf{A}_i + \mathbf{I})^{-1})(x_k - \Pi(x_k))$$

$$= \frac{1}{2} \alpha_k \gamma (x_k - \Pi(x_k))^T \mathbf{M}_k (x_k - \Pi(x_k)).$$

Therefore, (48) gives

$$\|x_{k+1} - \Pi(x_{k+1})\|^2 \leq \|x_k - \Pi(x_k)\|^2 - 2\alpha_k \gamma (x_k - \Pi(x_k))^{\top} \mathbf{M}_k (x_k - \Pi(x_k))$$

$$+ \frac{1}{2} \alpha_k \gamma (x_k - \Pi(x_k))^T \mathbf{M}_k (x_k - \Pi(x_k))$$

$$\leq \|x_k - \Pi(x_k)\|^2 - \frac{3}{2} \inf \alpha_k \gamma (x_k - \Pi(x_k))^T \mathbf{M}_k (x_k - \Pi(x_k)).$$

Taking full expectation and applying Lemma C.6, we obtain

$$\mathbb{E}\left[ \|x_{k+1} - \Pi(x_{k+1})\|^2 \right] \leq \left( 1 - \frac{3}{2} \inf \alpha_k \gamma \lambda_{\min}^+(\gamma) \right) \|x_k - \Pi(x_k)\|^2,$$

and hence

$$\mathbb{E}\left[ \|x_K - \Pi(x_K)\|^2 \right] \leq \left( 1 - \frac{3}{2} \inf \alpha_k \gamma \lambda_{\min}^+(\gamma) \right)^K \|x_0 - \Pi(x_0)\|^2.$$

$$\square$$

## G  NOTATION

| Notation | |
|---|---|
| $\|\cdot\|$ | Standard Euclidean norm |
| $\langle\cdot,\cdot\rangle$ | Standard Euclidean inner product |
| $[k]$ | $:= \{1,\ldots,k\}$ |
| $\mathbb{E}_k\left[\cdot\right]$ | Expectation conditioned on the first $k$ iterations |
| $n$ | Number of workers/nodes/clients/devices |
| $d$ | Dimensionality of the problem |
| $\gamma$ | Stepsize |
| $\alpha_k$ | Extrapolation parameter |
| $L, L_i, L_{\max}$ | Smoothness constants (Assumption 1.2) |
| $\mathrm{prox}_{\gamma f}(x)$ | Proximal operator of the function $f$ with parameter $\gamma$ (see (3)) |
| $M_f^\gamma$ | Moreau envelope of the function $f$ with parameter $\gamma$ (see (7)) |
| $M^\gamma$ | $:= \frac{1}{n}\sum_{i=1}^n M_{f_i}^\gamma$ |
| $L_\gamma$ | Smoothness constant of $M^\gamma$ |
| $\mu_\gamma^+$ | The smallest non-zero eigenvalue of the matrix $\nabla^2 M^\gamma$ (see Theorems 4.1, 5.1) / PŁ constant (see Theorems 7.2, 7.5) |
| $\pi(\gamma)$ | Total time per global iteration of Algorithm 1 / 2 |
| $T_\mu(\gamma)$ | Time complexity of Algorithm 1 |
| $T_\mu(\gamma, S)$ | Time complexity of Algorithm 2 |
| $\mathcal{X}_*$ | $:= \{x \in \mathbb{R}^d : \nabla f_i(x) = 0\}$ |
| $\Pi(x)$ | Projection of $x$ onto the solution set $\mathcal{X}_*$ |
| $R^2$ | $:= \|x_* - x_0\|^2$ |
| $D_f(x,y)$ | Bregman divergence between $x$ and $y$ associated with a function $f : \mathbb{R}^d \to \mathbb{R}$ |
| $\mathbf{I}$ | Identity matrix |
| $\mathbf{A}_i$ | Local data matrix stored on worker $i$, with spectral decomposition $\mathbf{A}_i = \mathbf{Q}_i \mathbf{\Lambda}_i \mathbf{Q}_i^\top$ |
| $\mathbf{A}$ | $:= \frac{1}{n}\sum_{i=1}^n \mathbf{A}_i$ |
| $\mathrm{Sym}_+^d$ | $:= \{\mathbf{X} \in \mathbb{R}^{d\times d} \,\vert\, \mathbf{X} = \mathbf{X}^\top, \mathbf{X} \succeq 0\}$ - the set of symmetric positive semidefinite matrices |
| $\ker \mathbf{A}$ | Kernel of a matrix $\mathbf{A}$ |
| $\mathbf{M}_k$ | $:= \frac{1}{\gamma}\left(\mathbf{I} - \frac{1}{S}\sum_{i\in\mathcal{S}_k}(\gamma\mathbf{A}_i + \mathbf{I})^{-1}\right)$ |
| $\mathbf{M}$ | $:= \frac{1}{\gamma}\left(\mathbf{I} - \frac{1}{n}\sum_{i=1}^n(\gamma\mathbf{A}_i + \mathbf{I})^{-1}\right)$ |
| $\lambda_{\max}(\mathbf{B})$ | The largest eigenvalue of matrix $\mathbf{B}$ |
| $\lambda_{\min}(\mathbf{B})$ | The smallest eigenvalue of matrix $\mathbf{B}$ |
| $\lambda_{\min}^+(\mathbf{B})$ | The smallest positive eigenvalue of matrix $\mathbf{B}$ |
| $[\mathbf{B}]_j$ | The $j$th diagonal element of a diagonal matrix $\mathbf{B}$ |
| $[b_j]_{jj}$ | A diagonal matrix with $b_j$ as the $j$th entry |

Table 1: Frequently used notation.

