# OpenReview forum: "Tighter Performance Theory of FedExProx"
_ICLR.cc/2025/Conference — Submitted to ICLR 2025_

### Official Review · Reviewer_kKTX · 2024-11-03

**Soundness:** 3
**Presentation:** 3
**Contribution:** 2
**Rating:** 5
**Confidence:** 4

**Summary:**

The authors re-examine a new approach to federated learning called FedExProx.
They analyze it in the case of convex quadratic objectives all while taking lags in computation and communication under consideration,
and show that it benefits standard approaches. This improves over previous analysis of the method.

**Strengths:**

1) The authors make a good job by taking into consideration also practical considerations like lags in communication and computation and reflecting them in the overall performance of the method

2) The authors illustrate a specific diagonal case where the improvement of what they suggest is obtained

**Weaknesses:**

1) The paper does not take the noise due to the samples into account: this is missing both in the convergence rates as well as not taken into account in
the cost of computing the prox operation

2) Focusing on quadratic functions is very limiting and does not properly reflect the more general case. Indeed, in the quadratic case, the gradients are linear in the weights which is a unique property that does not generalize

3) For quadratic functions, the authors should compare their approach to the FedAvg algorithm.
Noticelbly, the work of (Woodworth et al): (Is Local SGD Better than Minibatch SGD?) shows that for quadratic functions then FedAvg obtains the best possible convergence guarantees. So it is not clear what is the benefit of FedExProx on FedAvg in this case?

4) The extension to the case where the Moreu envelope is PL is nice, but it is not clear if it applies in any interesting case?

**Questions:**

) Please compare to FedAvg in the quadratic case, what do you get?

2) The parameter $L_{S,\gamma}$ is confusing to me, shouldn't it increase with $S$, similarly to randomized coordinate descent?

---

> ### Author Response · Authors · 2024-11-18
>
> First, we would like to thank the reviewer for thoughtful feedback. We will address each of the comments in detail.
>
> **Weaknesses**
>
> > The paper does not take the noise due to the samples into account: this is missing both in the convergence rates as well as not taken into account in the cost of computing the prox operation
>
> In the main part of this work, we focus on the clean setup where we assume access to an oracle that computes prox exactly, as the primary goal was to provide a theoretical explanation for the empirical performance gains observed when using extrapolation. We believe presenting the main results in this simplified setup is essential for clarity. However, the paper **does include an analysis of FedExProx with inexact computations**, which can be found in Appendix E.
>
> > Focusing on quadratic functions is very limiting and does not properly reflect the more general case. Indeed, in the quadratic case, the gradients are linear in the weights which is a unique property that does not generalize
>
> We do **not** restrict our analysis to quadratic problems alone. In Section 7, we extend our results to the general **general PŁ** case, a well-established and broadly applicable assumption [2] that encompasses, for example, all strongly convex functions.
>
> Extrapolation methods have long been heuristically known to improve convergence [1], yet until now, no work has demonstrated that they can be provably superior to the simplest baseline-Gradient Descent. Our work is **the first to rigorously establish that these algorithms can indeed outperform this baseline**. We consider this advancement an important contribution, as it provides a theoretical framework that fills a crucial gap in current understanding.
>
> Our analysis **significantly improves upon all previous results** and sheds new light on the method’s true performance. While our main focus is on quadratic problems, experimental evidence in our study, supported by prior research, suggests that the observed behavior extends beyond this specific context.
>
> Even if our work was limited to quadratic problems alone, we believe it would still be of value to the community. For instance, consider the paper [3], which only provides theoretical guarantees in the quadratic case and has nonetheless been a highly influential work that inspired extensive subsequent research. We believe that our results can similarly act as a foundation for future studies that relax our assumptions further and advance both the theory and practice of federated optimization.
>
> [1] Combettes, Patrick L.. “Hilbertian convex feasibility problem: Convergence of projection methods.” Applied Mathematics and Optimization 35 (1997): 311-330.
>
> [2] Karimi, H., Nutini, J. and Schmidt, M., 2016. Linear convergence of gradient and proximal-gradient methods under the Polyak-Łojasiewicz condition. In Machine Learning and Knowledge Discovery in Databases: European Conference, ECML PKDD 2016, Riva del Garda, Italy, September 19-23, 2016, Proceedings, Part I 16 (pp. 795-811). Springer International Publishing.
>
> [3] Shamir, O., Srebro, N. and Zhang, T., 2014, June. Communication-efficient distributed optimization using an approximate Newton-type method. In International conference on machine learning (pp. 1000-1008). PMLR.

---

> ### Author Response · Authors · 2024-11-18
>
> **Weaknesses (cont.)**
>
> > For quadratic functions, the authors should compare their approach to the FedAvg algorithm. Noticelbly, the work of (Woodworth et al): (Is Local SGD Better than Minibatch SGD?) shows that for quadratic functions then FedAvg obtains the best possible convergence guarantees. So it is not clear what is the benefit of FedExProx on FedAvg in this case?
>
> The referenced work demonstrates that **accelerated** FedAvg achieves optimal performance for quadratic functions in a **homogeneous** setting. In contrast, our study addresses the **heterogeneous** data scenario, where the findings of [1] do not apply. In our setting, the results in [2] demonstrate that heterogeneity can cause significant issues for FedAvg, leading to the well-known **client drift** problem. The paper proves a lower bound showing performance degradation with increasing heterogeneity, which implies that FedAvg is strictly worse than Minibatch SGD when heterogeneity is high, and thus **not optimal** (see also [3]). This is precisely why extrapolation techniques, such as those used in FedExProx, are beneficial.
>
> In fact, even the original analysis of FedExProx (which we improve upon) outperforms FedProx and FedExP, which in turn show improvements over FedAvg.
>
> We discuss this in the section "Related work" of our paper.
>
> [1] Woodworth, B., Patel, K.K., Stich, S., Dai, Z., Bullins, B., Mcmahan, B., Shamir, O. and Srebro, N., 2020, November. Is local SGD better than minibatch SGD?. In International Conference on Machine Learning (pp. 10334-10343). PMLR.
>
> [2] Karimireddy, S.P., Kale, S., Mohri, M., Reddi, S., Stich, S. and Suresh, A.T., 2020, November. Scaffold: Stochastic controlled averaging for federated learning. In International conference on machine learning (pp. 5132-5143). PMLR.
>
> [3] Woodworth, B.E., Patel, K.K. and Srebro, N., 2020. Minibatch vs local SGD for heterogeneous distributed learning. Advances in Neural Information Processing Systems, 33, pp.6281-6292.
>
> > The extension to the case where the Moreu envelope is PL is nice, but it is not clear if it applies in any interesting case?
>
> As noted in the paper, just below the introduction of Assumption 7.1, our assumption on Moreau envelopes **holds whenever the original function $f$ satisfies the PŁ condition**. The PŁ condition is a well-established assumption known to be quite general, and includes, for example, the whole class of strongly convex functions! An extensive description of the class of functions covered by the PŁ assumption, and thus by our Assumption 7.1, can be found in [1].
>
> [1] Karimi, H., Nutini, J. and Schmidt, M., 2016. Linear convergence of gradient and proximal-gradient methods under the Polyak-Łojasiewicz condition. In Machine Learning and Knowledge Discovery in Databases: European Conference, ECML PKDD 2016, Riva del Garda, Italy, September 19-23, 2016, Proceedings, Part I 16 (pp. 795-811). Springer International Publishing.
>
> **Questions:**
>
> > Please compare to FedAvg in the quadratic case, what do you get?
>
> Please, see the response above.
>
> > The parameter $L_{\gamma, S}$ is confusing to me, shouldn't it increase with $S$, similarly to randomized coordinate descent?
>
> The parameter $L_{\gamma, S}$ indeed decreases as $S$ increases, which is in fact quite intuitive. For instance, consider the iteration complexity presented in Theorem 5.1, given by $\mathcal{O}\left(\frac{L_{\gamma,S}}{\mu^{+}_{\gamma}} \log \frac{1}{\varepsilon}\right)$. This complexity is a decreasing function of $S$. As $S$ grows, more clients contribute to the training process, providing more information about the true descent direction. This increased participation results in fewer iterations needed for the method to converge.
>
> We thank the reviewer once again for their valuable suggestions. We believe that any potential adjustments are minor  and can be easily incorporated into the camera-ready version of the paper. We are happy to provide further details if needed.
> We hope these clarifications enhance the reviewer's appreciation of our results, as this is the first paper to theoretically explain an important phenomenon in FL, laying a foundation for future research. We kindly ask the reviewer to reconsider their score.

---

### Official Review · Reviewer_9m8x · 2024-11-04

**Soundness:** 3
**Presentation:** 3
**Contribution:** 2
**Rating:** 6
**Confidence:** 1

**Summary:**

This paper proposes a new analysis for FedExProx algorithm.

**Strengths:**

The analysis of the algorithm seems to be solid. It seems to improve over previous results. The authors also extended the results to partial participation scenarios.

**Weaknesses:**

The experiments only involves synthetic data.

**Questions:**

Have the authors considered running experiments on more realistic datasets?

---

> ### Author Response · Authors · 2024-11-18
>
> We would like to thank the reviewer for their feedback.
>
> We emphasize that the core of our work lies in the theoretical results rather than the experiments. While extrapolation methods have been heuristically shown to enhance convergence a long time ago [1], they have never been proven to be superior to the simplest baseline-Gradient Descent. Our work is **the first to rigorously demonstrate that such algorithms can indeed outperform this baseline**, making a significant contribution by bridging this knowledge gap. Our analysis **significantly improves upon all the previous results**, providing deeper insights into the method’s true performance.
>
> Although the experiments were not intended as the main focus, they are included to illustrate the robustness of our theoretical findings. Moreover, we are open to expanding the experimental section, including results on real data, for the camera-ready version of the paper.
>
> [1] Combettes, Patrick L.. “Hilbertian convex feasibility problem: Convergence of projection methods.” Applied Mathematics and Optimization 35 (1997): 311-330.

---

### Official Review · Reviewer_dS2L · 2024-11-05

**Soundness:** 3
**Presentation:** 3
**Contribution:** 2
**Rating:** 3
**Confidence:** 4

**Summary:**

This work revisited FedExProx, a recently proposed algorithm for FL, and furthered the convergence analysis. They first found the existing analysis of FedExProx is not as good as vanilla GD on quadratic problems. Then they provided a refined analysis for FedExProx on non-strongly convex quadratic problems regarding the time complexity, which beats GD. They also provided the analysis in partial participation, adaptive extrapolation strategy, and PL condition, and complete the story with numerical experiments.

**Strengths:**

1. Provided a refined analysis for FedExProx in some cases
2. Extend the analysis to more cases like partial participation and PL condition.
3. The paper is clearly written.

**Weaknesses:**

1. The scope of research is restricted to quadratic problems only, which may limit the significance of the work without enough motivation.
2. The claimed outperformance compared to GD (Theorem 4.3) is achieved with additional information on the communication protocol parameter $\mu$, computation protocol parameter $\tau$. I am skeptical that such setting are a bit coarse, I am not sure whether the communication parameter $\mu$ is easy to detect in the system, also $\tau$ corresponds to uniform computation cost for all devices, which may disregard the heterogeneity in computation among devices. Regarding authors claimed to study an empirically promising algorithm, I think authors should add more discussion on it for clarification, and to further improve the significance.
3. The stepsize choice in Theorem 6.1 depends on the exact value of the Moreau envelope or its lower bound, which is impractical in my opinion.
4. I suggest adding vanilla GD into your experiments to verify that your outperformance theory works in practice.

With that I think the work can be benefited a lot with a revision to further clarify the motivation and some technical details. Thank you.

**Questions:**

1. In your quadratic problems, I think the proximal operator attains a closed-form expression (as Eq(22) in Appendix C), why do you still use GD as the subroutine, rather than directly use the closed-form solution in Section 4.1?

---

> ### Author Response · Authors · 2024-11-18
>
> First, we would like to thank the reviewer for thoughtful feedback. We will address each of the comments in detail.
>
> **Weaknesses**
>
> > The scope of research is restricted to quadratic problems only, which may limit the significance of the work without enough motivation.
>
> We do **not** restrict ourselves to quadratic problems only. In Section 7 of the paper, we provide an analysis for the **general PŁ** case. The PŁ condition is a well-established assumption known to be quite general [2], and includes, for example, the whole class of strongly convex functions!
>
> While extrapolation methods have been heuristically shown to enhance convergence a long time ago [1], such methods have never been shown to be provably superior to the simplest baseline-Gradient Descent. Our work is **the first to rigorously prove that such algorithms can indeed outperform this baseline**. We believe that this advancement is an important contribution to the field, providing a theoretical framework that bridges a critical knowledge gap.
> Our analysis **significantly improves upon all the previous results**, offering insights into the true method’s performance. Although our primary focus is on quadratic problems, the experimental evidence presented in our work and supported by prior studies indicates that the observed phenomena extend beyond this specific setting.
>
> Even if our work had been limited to quadratic problems alone, we believe it would still offer valuable insights to the community. For instance, consider the paper [3], which only provides theoretical guarantees in the quadratic case and has nonetheless been a highly influential work that inspired extensive subsequent research. We believe that our results can similarly act as a stepping stone, encouraging future studies that relax our assumptions further and advance both the theory and practice of federated optimization.
>
> [1] Combettes, Patrick L.. “Hilbertian convex feasibility problem: Convergence of projection methods.” Applied Mathematics and Optimization 35 (1997): 311-330.
>
> [2] Karimi, H., Nutini, J. and Schmidt, M., 2016. Linear convergence of gradient and proximal-gradient methods under the Polyak-Łojasiewicz condition. In Machine Learning and Knowledge Discovery in Databases: European Conference, ECML PKDD 2016, Riva del Garda, Italy, September 19-23, 2016, Proceedings, Part I 16 (pp. 795-811). Springer International Publishing.
>
> [3] Shamir, O., Srebro, N. and Zhang, T., 2014, June. Communication-efficient distributed optimization using an approximate Newton-type method. In International conference on machine learning (pp. 1000-1008). PMLR.
>
> > The claimed outperformance compared to GD (Theorem 4.3) is achieved with additional information on the communication protocol parameter $\mu$, computation protocol parameter $\tau$. I am skeptical that such setting are a bit coarse, I am not sure whether the communication parameter $\mu$ is easy to detect in the system, also $\tau$ corresponds to uniform computation cost for all devices, which may disregard the heterogeneity in computation among devices.
>
> Let us unpack the result of Theorem 4.3. First, let us recall that when $\gamma \to 0$, then FedExProx effectively reduces to GD.
> Theorem 4.3 shows that when $\frac{\mu}{\tau}<2$ (i.e., communication time dominates computation time), then the optimal stepsize $\gamma^*$ belong to an interval that does not contain $0$.
> Crucially, one does **not** need to know $\mu$ and $\tau$ for this to be true. Regardless of what these parameters are, as long as $\frac{\mu}{\tau}<2$, FedExProx is provably better than GD. Detecting these parameters within the system is not required.
> Importantly, the condition $\frac{\mu}{\tau}<2$ typically holds in practice. In fact, the communication time can be massively larger than computation time. In federated learning scenarios, communication is typically the main bottleneck, since the clients are often limited in terms of communication bandwidth and links between the machines are slow [1]. This reinforces the practical relevance of our result in Theorem 4.3.
>
> While our analysis assumes uniform computation times across nodes, this assumption is common to all *synchronous* optimization methods, including the GD algorithm we benchmark against. In this context, $\tau$ can be viewed as an upper bound on the processing times of participating machines. Extending FedExProx to support asynchronous operations and account for client heterogeneity is an important research direction and deserves a separate work.
>
> [1] Kairouz, P., McMahan, H.B., Avent, B., Bellet, A., Bennis, M., Bhagoji, A.N., Bonawitz, K., Charles, Z., Cormode, G., Cummings, R. and D’Oliveira, R.G., 2021. Advances and open problems in federated learning. Foundations and trends in machine learning, 14(1–2), pp.1-210.

---

> ### Author Response · Authors · 2024-11-18
>
> **Weaknesses (cont.)**
>
> > Regarding authors claimed to study an empirically promising algorithm, I think authors should add more discussion on it for clarification, and to further improve the significance.
>
> As mentioned above, the empirical advantages of extrapolation have been demonstrated in prior research, which is why we identify the algorithm as empirically promising and aim to understand the underlying reasons for its performance.
>
> > The stepsize choice in Theorem 6.1 depends on the exact value of the Moreau envelope or its lower bound, which is impractical in my opinion.
>
> We assume that the workers have access to both the function and gradient values of the Moreau envelopes following previous works on proximal methods.
>
> The knowledge of lower bounds on the Moreau envelopes (which by Lemma A.5 is equivalent to knowing the lower bounds on the functions $f_i$) is required only in the Polyak-like step size rule (FedExProx-StoPS). For ML problems, this is a very weak assumption because most losses (hinge loss, log loss, euclidean loss) have lower bounds equal to $0$.
> This knowledge is the price we have to pay for not having to know $\gamma$-as discussed in Remark 6.3, FedExProx-StoPS converges at the same rate as that in Theorem 4.1 for *any* choice of $\gamma$. Hence, the method from Theorem 6.1 requires the knowledge of $\inf M_{f_i}^{\gamma}$, but the knowledge of $\alpha = \frac{1}{\gamma L_{\gamma}}$ from Theorem 4.1 is not needed. Depending on prior knowledge, one can select between different methods.
>
> Note that in Section E, we explain how one can use **inexact proxes** in FedExProx.
>
> > I suggest adding vanilla GD into your experiments to verify that your outperformance theory works in practice.
>
> Please, note that our experiments do cover the vanilla GD case, since FedExProx with $\gamma \to 0$ reduces to GD.
>
> **Questions**
>
> > In your quadratic problems, I think the proximal operator attains a closed-form expression (as Eq(22) in Appendix C), why do you still use GD as the subroutine, rather than directly use the closed-form solution in Section 4.1?
>
> The reason for using a solver is motivated by practical considerations. It is true that in this case there exist a closed form solution which can be easily computed for small $d$. However, in practical larger-scale problems, this is no longer the case, as inverting large matrices becomes prohibitively expensive. Therefore, one usually resorts to some iterative method. In this work, we consider GD as an example, but our theory could be extended to any other iterative solver.
>
>
> We again thank the reviewer for all the valuable suggestions. We are open to providing more details and believe that any adjustments can be easily incorporated into the camera-ready version of the paper.
> We hope that with these clarifications, the reviewer will appreciate our results. Let us reiterate that this is the first paper to theoretically explain an important phenomenon in FL, laying the groundwork for further research. We kindly ask the reviewer to reconsider their score.

---

### Author Response · Authors · 2024-11-24

Dear Reviewers,

We have addressed your questions and provided detailed responses to clarify all the concerns raised. May we kindly ask you to reply to the rebuttal? We would greatly appreciate your opinion.
Thank you for your time and consideration.

Best regards,
Authors

---

### Meta-Review · Area_Chair_9fRU · 2024-12-26

**Metareview:**

The paper presents notable theoretical advancements in federated optimization, particularly through its enhanced analysis of FedExProx. The theoretical contributions are seen as valuable additions to the field. The reviewers mostly concerned with the limited setting (quadratic case) which is clarified by the authors.  However, the paper has significant shortcomings in its empirical validation. The reviewers point out that the theoretical claims need experimental support, specifically through direct comparisons with GD method. Additionally, the paper lacks sufficient discussion of real-world implementation challenges, particularly regarding device heterogeneity in federated learning environments. Overall, while the theoretical foundation is strong, the work needs more empirical support and practical context to maximize its impact in the field.

**Additional Comments On Reviewer Discussion:**

The reviewers suggest that a revised version should include comprehensive experimental validation of the theoretical findings and incorporate a thorough analysis of practical implementation considerations. These additions would substantially strengthen the paper's contribution and make it more relevant for real-world applications in federated learning.

---

### Decision · Program_Chairs · 2025-01-22

Reject